# Flow Connecting Actions and Reactions: A Condition-Free Framework for Human Action-Reaction Synthesis

## Abstract

Human action-reaction synthesis, a fundamental challenge in modeling causal human interactions, plays a critical role in applications ranging from virtual reality to social robotics. While diffusion-based models have demonstrated promising performance, they exhibit two key limitations for interaction synthesis: reliance on complex noise-to-reaction generators with intricate conditional mechanisms, thus limiting to unidirectional generation, and frequent physical violations in generated motions. To address these issues, we propose Action-Reaction Flow Matching (ARFlow), a novel paradigm that establishes direct action-to-reaction mappings, eliminating the need for complex conditional mechanisms and supporting bi-directional generation. Directly applying traditional guidance algorithms tends to undermine the quality of generated reaction motion. We analyze the sampling of flow matching in depth and reveal an issue (Initial Point Deviation) which causes the sampling trajectory to ever farther from the initial action motion. Thus, we propose a reprojection guidance method, Re-Guid, to correct this deviation to enable better interaction. To further enhance the reaction diversity, we incorporate randomness into the sampling process. Extensive experiments on NTU120, Chi3D and InterHuman datasets demonstrate that ARFlow not only outperforms existing methods in terms of Fréchet Inception Distance and motion diversity but also significantly reduces body collisions, as measured by our introduced Intersection Volume and Intersection Frequency metrics.

## 1 Introduction

Human action-reaction synthesis (Tan et al.; Chopin et al., 2023) has emerged as a pivotal research direction in computer vision (Starke et al., 2020; Javed et al., 2024; Tanaka & Fujiwara, 2023; Wang et al., 2023). This task aims to generate physically plausible human reactions responding to observed actions, with critical applications in virtual reality, human-robot interaction, and character animation. Unlike single-human motion generation (Guo et al., 2020; Chen et al., 2023), reactors must infer responses without observing future actor motions, creating unique modeling challenges.

While recent diffusion methods (Tevet et al., 2023) show promise in motion generation, they face two key limitations in the modeling of action-reaction interactions. First, existing approaches (Xu et al., 2024) indirectly model responses using noise-to-reaction generators with intricate conditional mechanisms like treating action information as a condition to guide the generation process. This not only complicates the training process but also limits to unidirectional generation(action-to-reaction), which makes it completely fail in the interaction where the roles of actor and reactor continuously switch. Second, frequent physical violations like body penetration between characters occur due to neglected physical constraints. While such issues are absent in single-human scenarios, they become critical in human interaction applications (Hoyet et al., 2012). This poses a significant barrier to real-world applications such as virtual reality and human-robot interaction, where even minor physical inaccuracies are intolerable (Reitsma & Pollard, 2003; Hoyet et al., 2012).

To address these challenges, we propose Action-Reaction Flow Matching (**ARFlow**), a novel framework that fundamentally resolves these limitations. Unlike diffusion models constrained by noise-data mappings, flow matching (Lipman et al., 2023; Liu et al., 2023a) naturally models paired distributions

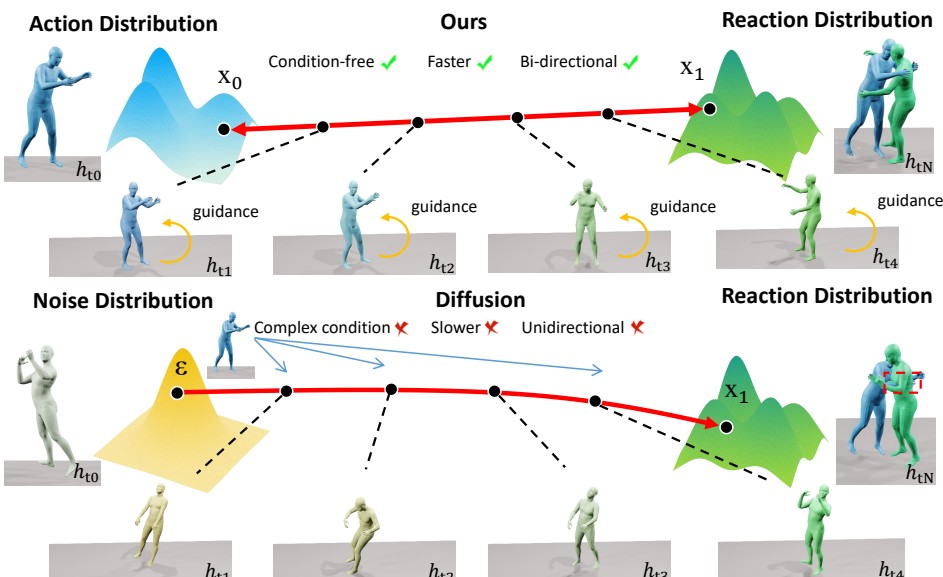

Figure 1: Our proposed Human Action-Reaction Flow (**ARFlow**). We directly establish a mapping between the action and reaction distribution and our sampling process is further guided by our reprojection guidance method (**RE-GUID**). The change of colors represents the variation of the $h$-frame human reaction mesh with respect to sampling timestep $t_n$.

through linear interpolation between endpoints (See Fig. 1), enabling simpler training and faster inference. Due to the establishment of direct pathways between action and reaction distributions, ARFlow **eliminates design of conditions**, thus supporting bi-directional generation of actions and reactions. To eliminate unrealistic body collisions between characters, traditional guidance algorithms (Karunratanakul et al., 2024; Li et al., 2024) tend to undermine the quality of generated reaction motions. We analyze the sampling of flow matching in depth and discover the issue of Initial Point Deviation. Thus, we propose a reprojection guidance method, **RE-GUID**, to correct this deviation to enable better interaction. This innovation maintains physical plausibility through gradient guidance without compromising motion quality. Our main contributions are as follows:

- We propose ARFlow, the first flow matching architecture that creates direct pathways between human action and reaction distributions, eliminating the design of conditions and supporting bi-directional generation compared to existing diffusion-based methods.

- We reveal an issue, *initial point deviation*, that occurred during sampling when flow matching models the distribution of actions and reactions. Flow matching sampling actually interpolates back towards the predicted mean point of the source distribution instead of the true initial point, and the accumulating bias pulls the trajectory ever farther from the expected start.

- We propose a reprojection guidance method, RE-GUID, to correct the *initial point deviation* to enable better interaction. Our reprojection guidance method does not require differentiating the neural network, further improving efficiency. Moreover, we propose using a weighted direction of random direction and sampling direction during the sampling process to support diverse reaction motions for the same action.

## 2 RELATED WORK

**Human Action-Reaction Synthesis.** Different from human-human interaction (Liang et al., 2023; Starke et al., 2020; Javed et al., 2024; Wang et al., 2023), human action-reaction synthesis is causal and asymmetric (Liu et al., 2019; Xu et al., 2023). To address this task, researchers have leveraged large language models (Siyao et al., 2024; Tan et al.; Jiang et al., 2024), VAE-based methods (Chopin et al., 2023; Liu et al., 2023b; 2024). However, these methods cannot capture fine-grained representations and ensure diversity, and diffusion-based methods (Li et al., 2024; Tanaka & Fujiwara, 2023) are

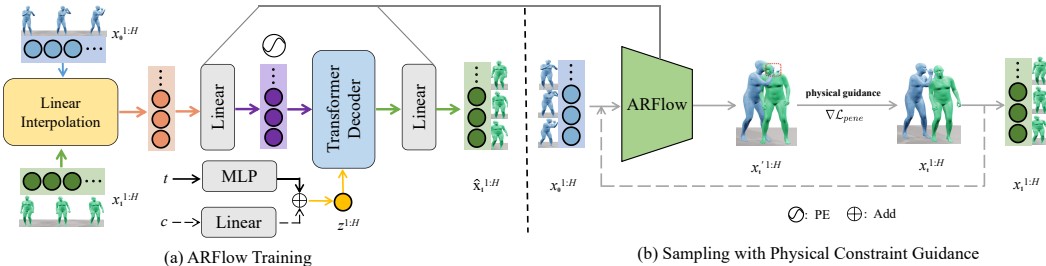

(a) ARFlow Training

(b) Sampling with Physical Constraint Guidance

Figure 2: **Pipeline of ARFlow.** **(a)** For a sampled timestep $t$, we linearly interpolate a coupled action-reaction pair as Eq. 4 to produce the intermediate state $x_t^{1:H}$, which is then turns into a $d$-dimensional latent feature through a linear layer. We use Transformer Decoder Units to directly predict clean reaction motions. **(b)** After training the networks in (a), ARFlow sampling is further guided by our reprojection guidance method (RE-GUID) to generate physically plausible reactions.

limited to the "offline" and "constrained" setting of human reaction generation, failing to generate instant and intention agnostic reactions. More recently, ReGenNet (Xu et al., 2024) introduce a diffusion-based transformer decoder framework and treat action sequence as conditional signal for online reaction generation. However, it often produce physically-implausible inter-penetrations between the actor and reactor since they disregard physical constraints in the generative process. Our method addresses this problem by ARFlow sampling with physical constraint guidance.

**Flow Matching.** Flow Matching (Lipman et al., 2023; Liu et al., 2023a; Martin et al., 2024; Feng et al., 2025) has emerged as an efficient alternative to diffusion models, offering linear generation trajectories through ODE solvers. This paradigm enables simplified training and accelerated inference (Lipman et al., 2024), with successful applications spanning images (Esser et al., 2024), audio (Le et al., 2023), video (Aram Davtyan & Favaro, 2023), and point clouds (Wu et al., 2023). In motion generation, MotionFlow (Hu et al., 2023) demonstrates comparable performance to diffusion models with faster sampling. Notably, Flow Matching inherently models transitions between arbitrary distributions through transport maps, making it particularly suitable for paired data modeling. Despite these advantages, its potential for action-reaction synthesis remains unexplored. Our work bridges this gap by establishing direct action-to-reaction mappings without complex conditional mechanisms.

## 3 METHOD

In the setting of human action-reaction synthesis, our primary goal is to generate the reaction $\mathbf{x}_1 = \{x_1^i\}_{i=1}^H$ conditioned on an arbitrary action $\mathbf{x}_0 = \{x_0^i\}_{i=1}^H$ of length $H$. The condition $\mathbf{c}$ can be action $\mathbf{x}_0$, or it can be a signal such as an action label, text, audio to instruct the interaction, which is optional for intention-agnostic scenarios. We utilize SMPL-X (Pavlakos et al., 2019) human model to represent the human motion sequence as Xu et al. (2024) to improve the modeling of human-human interactions. Thus, the reaction can be represented as $x_1^i = [\boldsymbol{\theta}_i^{x_1}, \boldsymbol{q}_i^{x_1}, \boldsymbol{\gamma}_i^{x_1}]$ where $\boldsymbol{\theta}_i^{x_1} \in \mathbb{R}^{3K}$, $\boldsymbol{q}_i^{x_1} \in \mathbb{R}^3$, $\boldsymbol{\gamma}_i^{x_1} \in \mathbb{R}^3$ are the pose parameters, the global orientation, and the root translation of the person, respectively. Total number $K$ of body joints, including the jaw, eyeballs, and fingers, is 54. The main pipeline of our ARFlow model is provided in Fig. 2. In this section, we first introduce the Human Action-Reaction Flow Matching in Sec. 3.1. Then, we present our reprojection guidance method to address the issue of physically implausible human-human inter-penetrations in Sec. 3.2.

### 3.1 HUMAN ACTION-REACTION FLOW MATCHING

**Flow Matching Overview.** Given a set of samples from an unknown data distribution $q(\mathbf{x})$, the goal of flow maching is to learn a *flow* that transforms a prior distribution $p_0(\mathbf{x})$ towards a target data distribution $p_1(\mathbf{x}) \approx q(\mathbf{x})$ along the probability path $p_t(\mathbf{x})$. The time-dependent flow $\phi_t(\mathbf{x})$ is defined by a vector field $\mathbf{v}(\mathbf{x}, t) : \mathbb{R}^d \times [0, 1] \to \mathbb{R}^d$ which establishes the flow through a neural ODE:

$$\frac{d}{dt}\phi_t(\mathbf{x}) = \mathbf{v}(\phi_t(\mathbf{x}), t), \qquad \phi_0(\mathbf{x}) = \mathbf{x} . \tag{1}$$

Given a predefined probability path $p_t(\mathbf{x})$ and a corresponding vector field $\mathbf{u}_t(\mathbf{x})$, one can regress the vector field $\mathbf{u}_t(\mathbf{x})$ with a neural network $\mathbf{v}_\theta(\mathbf{x}_t, t)$ parameterized by $\theta$, and the Flow Maching (FM) objective is as follows:

$$\min_\theta \mathbb{E}_{t, p_t(\mathbf{x})} \|\mathbf{v}_\theta(\mathbf{x}_t, t) - \mathbf{u}_t(\mathbf{x})\|^2. \tag{2}$$

By defining the conditional probability path as a linear interpolation between $p_0$ and $p_1$, the intermediate process becomes: $\mathbf{x}_t = t\mathbf{x}_1 + [1 - (1 - \sigma_{\min})t]\mathbf{x}_0$, where $\sigma_{\min} > 0$ is a small amount of noise. Both training and sampling are simplified by fitting a linear trajectory in contrast to diffusion paths. When extra condition signals $\mathbf{c}$ are required, they can be directly incorporated into the vector field estimator $\mathbf{v}_\theta(\mathbf{x}_t, t)$ as $\mathbf{v}(\mathbf{x}_t, t, \mathbf{c})$. Therefore, the training objective is as follows:

$$\min_\theta \mathbb{E}_{t, p(\mathbf{x}_0), q(\mathbf{x}_1)} \left\| \mathbf{v}_\theta(\mathbf{x}_t, t, c) - \left( \mathbf{x}_1 - (1 - \sigma_{\min})\mathbf{x}_0 \right) \right\|^2. \tag{3}$$

Since $\mathbf{c}$ is an optional action label in this task and is an empty value on our unconstrained experimental settings, we can ignore it in the following text.

**Action-Reaction Flow Matching.** Different from previous diffusion-based methods(Xu et al., 2024; Tevet et al., 2023; Li et al., 2024; Du et al., 2023) that rely on cumbersome conditional mechanisms, we adopt flow matching to directly construct a mapping from action distribution to reaction distribution (See Fig. 1). Specially, we build a **condition-free** generative model $f$ parametrized by $\theta$ to synthesize the reaction $\mathbf{x}_1 = f_\theta(\mathbf{x}_0)$, given action $\mathbf{x}_0$, instead of $\mathbf{x}_1 = f_\theta(\mathbf{z}, \mathbf{y})$ in diffusion, given a sampled Gaussian noise vector $\mathbf{z}$ and an action motion $\mathbf{y}$ as a condition. Due to the elimination of conditional design by directly constructing the ODE trajectories of actions and reactions through Eq. 1, ARFlow enables **bi-directional** generation, i.e., the inversion of action-to-reaction models can serve as reaction-to-action models to support the interaction (e.g., boxing) where the roles of actor and reactor **continuously switch**. In this scenario, we also need the model to be able to perform reverse generation (reaction-to-action) which diffusion-based methods cannot achieve. Given the reaction $\mathbf{x}_1$ sampled from the reaction distribution and the coupled action $\mathbf{x}_0$ from the action distribution, the intermediate process can be written as

$$\mathbf{x}_t = t\mathbf{x}_1 + [1 - (1 - \sigma_{\min})t]\mathbf{x}_0, \tag{4}$$

where $t$ is the timestep, $\sigma_{\min} > 0$ is a small amount of noise. In our setting, our samples are drawn from the marginal distribution $p(\mathbf{x}_{t_{n+1}} | \mathbf{x}_{t_n})$ rather than the conditional distribution $p(\mathbf{x}_{t_{n+1}} | \mathbf{x}_{t_n}, \mathbf{y})$. We use a neural network $G$ to directly predict the clean body poses, *i.e.*, $\hat{\mathbf{x}}_1 = G_\theta(\mathbf{x}_t, t)$, instead of predicting vector fields in previous works (Hu et al., 2023; Lipman et al., 2023). This strategy is both straightforward and effective, since many geometric losses directly act on the predicted $\hat{\mathbf{x}}_1$. We compared and analyzed the results of predicting vector fields ($\mathbf{v}$-prediction) and clean body poses ($\mathbf{x}_1$-prediction) in Sec. 4.3. Note that $\mathbf{x}_1$ in flow maching usually corresponds to $\mathbf{x}_0$ in previous literature on diffusion models. Depending on the specific application, $G$ can be implemented by Transformers (Vaswani et al., 2017) or MLP networks. The training objective of our flow model is as follows:

$$\mathcal{L}_{\text{fm}} = \mathbb{E}_{\mathbf{x}_1 \sim q(\mathbf{x}_1), \mathbf{x}_0 \sim p(\mathbf{x}_0), t \sim [0,1]}[\|\mathbf{x}_1 - G_\theta(\mathbf{x}_t, t)\|_2^2]. \tag{5}$$

Following Xu et al. (2024), we employ explicit interaction losses to evaluate the relative distances of body pose $\boldsymbol{\theta}(\mathbf{x}_1, \mathbf{x}_0)$, orientation $\boldsymbol{q}(\mathbf{x}_1, \mathbf{x}_0)$ and translation $\boldsymbol{\gamma}(\mathbf{x}_1, \mathbf{x}_0)$ between the actor and reactor. We use a forward kinematic function to transforms the rotation pose into joint positions for calculating $\boldsymbol{\theta}(\mathbf{x}_1, \mathbf{x}_0)$, and converts the rotation poses to rotation matrices for calculating $\boldsymbol{q}(\mathbf{x}_1, \mathbf{x}_0)$. The interaction loss is defined as

$$\mathcal{L}_{\text{inter}} = \frac{1}{H} \bigg( \|\boldsymbol{\theta}(\mathbf{x}_1, \mathbf{x}_0) - \boldsymbol{\theta}(\hat{\mathbf{x}}_1, \mathbf{x}_0)\|_2^2$$

$$+ \|\boldsymbol{q}(\mathbf{x}_1, \mathbf{x}_0) - \boldsymbol{q}(\hat{\mathbf{x}}_1, \mathbf{x}_0)\|_2^2 + \|\boldsymbol{\gamma}(\mathbf{x}_1, \mathbf{x}_0) - \boldsymbol{\gamma}(\hat{\mathbf{x}}_1, \mathbf{x}_0)\|_2^2 \bigg). \tag{6}$$

Our overall training loss is $\mathcal{L}_{\text{all}} = \mathcal{L}_{\text{fm}} + \lambda_{\text{inter}} \cdot \mathcal{L}_{\text{inter}}$, and $\lambda_{\text{inter}}$ is the loss weight.

**Sampling based on $\mathbf{x}_1$-prediction.** Since our neural network outputs $\hat{\mathbf{x}}_1$, we require to construct an equivalent relationship between the neural network's predictions of $\mathbf{v}$ and $\mathbf{x}_1$. The equivalent form of parameterization Eq. 21 derived from our appendix is as follows:

$$\mathbf{v}_\theta(\mathbf{x}_t, t, c) = \frac{\hat{\mathbf{x}}_1 - (1 - \sigma_{\min})\mathbf{x}_t}{1 - (1 - \sigma_{\min})t}, \tag{7}$$

Then, our sampling based on $\mathbf{x}_1$-prediction can be achieved by first sampling $\mathbf{x}_0$ and then solving Eq. 1 employing an ODE solver (Runge, 1895; Kutta, 1901; Alexander, 1990) through our trained neural network $G_\theta$. We use the Euler ODE solver and discretization process involves dividing the procedure into $N$ steps, leading to the following formulation:

$$\mathbf{x}_{t_{n+1}} \leftarrow \mathbf{x}_{t_n} + (t_{n+1} - t_n)\,\mathbf{v}_\theta(\mathbf{x}_{t_n}, t_n, \mathbf{c}), \tag{8}$$

where the integer time step $t_1 = 0 < t_2 < \cdots < t_N = 1$. By using equivalent form of parameterization Eq. 7, we finally obtain our flow maching sampling formulation based on $\mathbf{x}_1$-prediction:

$$\mathbf{x}_{t_{n+1}} \leftarrow \frac{1 - (1 - \sigma_{\min})t_{n+1}}{1 - (1 - \sigma_{\min})t_n}\mathbf{x}_{t_n} + \frac{t_{n+1} - t_n}{1 - (1 - \sigma_{\min})t_n}\,\hat{\mathbf{x}}_1, \tag{9}$$

which is more suitable for human motion generation. Detailed derivation is provided in **Appendix** A. However, traditional flow matching sampling is deterministic and cannot generate diverse reaction motions for the same action. We address this issue in Sec. 3.2.

## 3.2 RE-GUID: REPROJECTION GUIDANCE METHOD

To address physically implausible inter-penetrations between the actor and reactor in the generated results of current diffusion-based methods (Xu et al., 2024; Tevet et al., 2023; Du et al., 2023), traditional guidance methods (Karunratanakul et al., 2024; Li et al., 2024) employ a penetration gradients $\nabla \mathcal{L}_{\text{pene}}$ to guide the sampling process. The penetration loss functon to calculate the signed distance function (SDF) between the actor and the reactor is as follows

$$\mathcal{L}_{\text{pene}}(\mathbf{x}) := \sum_{i,h} -\min\Big(\text{SDF}(\psi_i^h(\mathbf{x})), \zeta\Big), \tag{10}$$

where $\psi_i^h(\mathbf{x})$ represents the position of the $i$-th joint in the $h$-th frame of the generated reaction motion $\mathbf{x}$, the $\zeta$ defines the safe distance between the actor and the reactor, beyond which the gradient becomes zero, and SDF is the signed distance function for an actor in the $h$-th frame, which dynamically changes across frames.

However, these methods (Chung et al., 2023; Karunratanakul et al., 2023; Tian et al., 2024) first estimate $\hat{\mathbf{x}}_1$ from current state $\mathbf{x}_{t_n}$ with a denoiser network $\epsilon_\theta(\mathbf{x}_{t_n}, t_n, \mathbf{c})$, and then calculate gradients of the loss function with respect to current state $\mathbf{x}_{t_n}$, so it inevitably requires differentiation of the neural network, resulting in inaccurate gradients $\nabla \mathcal{L}_{\text{pene}}$.

**Initial Point Deviation.** Except for inaccurate gradients, since we build flow matching between the action and reaction distribution, our source distribution is the action distribution instead of noise distribution. Thus, we cannot simply add noise back like diffusion. However, the traditional flow matching sampling algorithm Eq. 9 is equivalent to the following formulation:

$$\hat{\mathbf{x}}_0 \leftarrow \hat{\mathbf{x}}_1 + \frac{\mathbf{x}_{t_n} - (1 + \sigma_{\min}t_n)\hat{\mathbf{x}}_1}{1 - (1 - \sigma_{\min})t_n}, \tag{11}$$

$$\mathbf{x}_{t_{n+1}} \leftarrow t_{n+1}\hat{\mathbf{x}}_1 + [1 - (1 - \sigma_{\min})t_{n+1}]\,\hat{\mathbf{x}}_0. \tag{12}$$

This sampling process essentially finds a $\hat{\mathbf{x}}_0$ along the opposite direction of the current velocity field for linear interpolation as Eq. 11 (black dotted lines in Fig. 3). Obviously, this predicted $\hat{\mathbf{x}}_0$ deviates from the initial point $\mathbf{x}_0$ (purple dotted lines in Fig. 3). In fact, this predicted $\hat{\mathbf{x}}_0$ is the mean of the source distribution learned by the neural network. Because this mean point rarely coincides with the actual initial point, there is a deviation in the interpolation direction from the beginning. During the sampling process, this bias accumulates, causing the trajectory to increasingly deviate from the expected starting state $\mathbf{x}_0$.

**Reprojection guidance.** To address these issues, we propose **RE-GUID**, that **first** directly updates the gradient at $\hat{\mathbf{x}}_1$ to avoid differentiation of the neural network:

$$\hat{\mathbf{x}}_1' \leftarrow \hat{\mathbf{x}}_1 - \lambda_{\text{pene}} \nabla_{\hat{\mathbf{x}}_1} \mathcal{L}_{\text{pene}}(\hat{\mathbf{x}}_1), \tag{13}$$

where $\hat{\mathbf{x}}_1$ is the clean body poses predicted by our neural network $G_\theta$ and $\lambda_{\text{pene}}$ is the guidance strength. **Then**, we use the linear interpolation of flow matching to **reproject back** to the intermediate

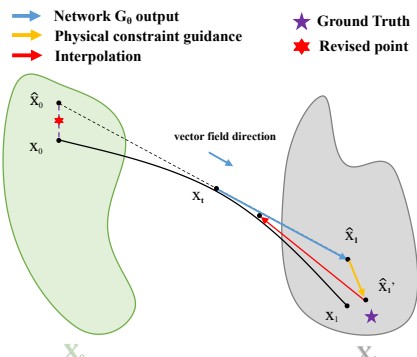

Figure 3: Illustration of Initial Point Deviation and our guidance method (RE-GUID).

state of learned FM path. In order to correct the projection direction of traditional flow matching, we use a weight factor $w$ to weight $\hat{\mathbf{x}}_0$ and $\mathbf{x}_0$:

$$\hat{\mathbf{x}}_0^* \leftarrow w\hat{\mathbf{x}}_0 + (1 - w)\mathbf{x}_0. \tag{14}$$

and use $\hat{\mathbf{x}}_0^*$ as our final endpoint for interpolation. Our reprojection guidance method RE-GUID and traditional guidance algorithm for $\mathbf{x}_1$-prediction are shown in Algorithm 2 and 1 respectively. In practice, we use $\mathbf{x}_1$-prediction for its better performance. Under iterative sampling and physical constraint guidance, our method can generate more realistic and physically-plausible reaction motions.

Our guidance method is actually a refined fine-tuning, which may be not suitable for training. In addition, the loss during the training mainly measures the difference between generated results and ground truth, while our guidance during the inference phase can provide more flexible guidance based on the quality of the generated results.

**Stochastic sampling to enhance diversity of reactions.** To generate diverse reaction motions for the same action, we incorporate randomness into the sampling process. The interpolation Eq. 12 can be written in the following equivalent form:

$$\mathbf{x}_{t_{n+1}} \leftarrow \hat{\mathbf{x}}_1 + (1 - t_{n+1})(\hat{\mathbf{x}}_0 - \hat{\mathbf{x}}_1) + \sigma_{\min}t_{n+1}\hat{\mathbf{x}}_0. \tag{15}$$

The interpolation process can be understood as a projection in the opposite direction of the current learned velocity field $\hat{\mathbf{x}}_1 - \hat{\mathbf{x}}_0$. Thus, we can weight the projection direction $\hat{\mathbf{x}}_0^* - \hat{\mathbf{x}}_1'$ and stochastic direction $d_{\mathrm{random}}$ to incorporate randomness:

$$d_{\mathrm{mix}} \leftarrow \hat{\mathbf{x}}_0^* - \hat{\mathbf{x}}_1' + \beta[d_{\mathrm{random}} - (\hat{\mathbf{x}}_0^* - \hat{\mathbf{x}}_1')], \tag{16}$$

$$\mathbf{x}_{t_{n+1}} \leftarrow \hat{\mathbf{x}}_1' + (1 - t_{n+1})d_{\mathrm{mix}} + \sigma_{\min}t_{n+1}\hat{\mathbf{x}}_0^*, \tag{17}$$

where $\beta$ is the factor to control the strength of randomness.

## 4 EXPERIMENTS

Our experiment setting of human action-reaction synthesis is **online** and **unconstrained** as in Xu et al. (2024) for its significant potential for practical applications. **Online** represents real-time reaction generation where future motions of the actor are not visible to the reactor, and the opposite is **offline** to relax the synchronicity. **Unconstrained** means that the intention of the actor is invisible to the reactor. To demonstrate the universality of our method, we also conducted offline setting experiments.

### 4.1 EXPERIMENT SETUP

**Evaluation Metrics.** 1) We adopt the following metrics to quantitatively evaluate results: Frechet Inception Distance (FID), Action Recognition Accuracy (Acc.), Diversity (Div.) and Multi-modality (Multimod.). For all these metrics widely used in previous human motion generation (Guo et al., 2020; Petrovich et al., 2021; Tevet et al., 2023; Xu et al., 2024), we use the action recognition model (Yan et al., 2018) to extract motion features for calculating these metrics as in Xu et al. (2024). We generate 1,000 reaction samples by sampling actor motions from test sets and evaluate each method 20 times using different random seeds to calculate the average with the 95% confidence interval as prior works (Guo et al., 2020; Petrovich et al., 2021; Tevet et al., 2023; Xu et al., 2024).

**Algorithm 1** Traditional guidance method of physical constraints.

1: **Input**: $\mathcal{L}_{\text{pene}}$ the loss function ; $G$ and $\theta$ the clean body poses predictor with pretrained parameters
2: **Parameters**: $N$ the number of sampling steps; $\lambda_{\text{pene}}$ the guidance strength
3: Sample $\mathbf{x}_0$ from the action distribution
4: **for** $n = 1, 2, ..., N - 1$ **do**
5: $\quad \hat{\mathbf{x}}_1 \leftarrow G_\theta(\mathbf{x}_{t_n}, t_n, \mathbf{c})$
6: $\quad$ # Flow Matching $\mathbf{x}_1$-prediction sampling (Eq. 9)
7: $\quad \mathbf{x}'_{t_{n+1}} \leftarrow \frac{1-t_{n+1}}{1-t_n}\mathbf{x}_{t_n} + \frac{t_{n+1}-t_n}{1-t_n}\hat{\mathbf{x}}_1$
8: $\quad$ # Physical constraint guidance
9: $\quad \mathbf{x}_{t_{n+1}} \leftarrow \mathbf{x}'_{t_{n+1}} - \lambda_{\text{pene}}\nabla_{\hat{\mathbf{x}}_{t_n}}\mathcal{L}_{\text{pene}}(\hat{\mathbf{x}}_1)$
10: **end for**
11: **Return**: The reaction motion $\mathbf{x}_1 = \mathbf{x}_{t_N}$

**Algorithm 2** Our reprojection guidance method (RE-GUID).

1: **Input**: $\mathcal{L}_{\text{pene}}$ the loss function ; $G$ and $\theta$ the clean body poses predictor with pretrained parameters
2: **Parameters**: $N$ the number of sampling steps; $\lambda_{\text{pene}}$ the guidance strength; $w$ weight factor
3: Sample $\mathbf{x}_0$ from the action distribution
4: **for** $n = 1, 2, ..., N - 1$ **do**
5: $\quad \hat{\mathbf{x}}_1 \leftarrow G_\theta(\mathbf{x}_{t_n}, t_n, \mathbf{c})$
6: $\quad \hat{\mathbf{x}}_0 \leftarrow \hat{\mathbf{x}}_1 + \frac{\mathbf{x}_{t_n} - (1+\sigma_{\min}t)\hat{\mathbf{x}}_1}{1-(1-\sigma_{\min})t_n}$ $\quad$ # (Eq. 11)
7: $\quad$ # Physical constraint guidance at $\hat{\mathbf{x}}_1$(Eq. 13)
8: $\quad \hat{\mathbf{x}}'_1 \leftarrow \hat{\mathbf{x}}_1 - \lambda_{\text{pene}}\nabla_{\hat{\mathbf{x}}_1}\mathcal{L}_{\text{pene}}(\hat{\mathbf{x}}_1)$
9: $\quad$ # Direction correction
10: $\quad \hat{\mathbf{x}}_0^* \leftarrow w\hat{\mathbf{x}}_0 + (1-w)\mathbf{x}_0$
11: $\quad$ # Interpolation (Eq. 12)
12: $\quad \mathbf{x}_{t_{n+1}} \leftarrow t_{n+1}\hat{\mathbf{x}}'_1 + [1 - (1-\sigma_{\min})t_{n+1}]\hat{\mathbf{x}}_0^*$
13: **end for**
14: **Return**: The reaction motion $\mathbf{x}_1 = \mathbf{x}_{t_N}$

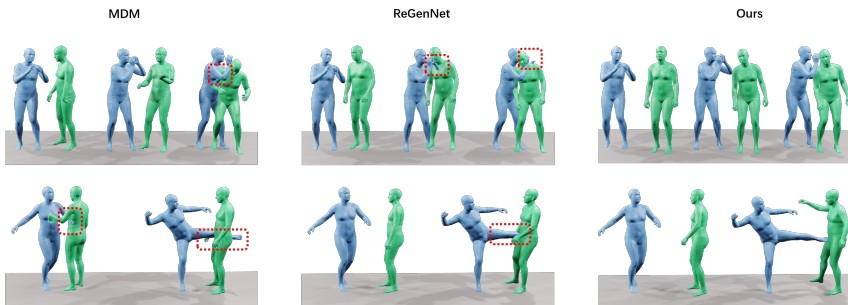

Figure 4: **Qualitative comparisons** of human action-reaction synthesis results. Blue for actors and Green for reactors.

To qualitatively measure the degree of penetration, we introduced two metrics: 2) **Intersection Volume (IV)** measures human-human inter-penetration by reporting the volume occupied by two human meshes. 3) **Intersection Frequency (IF)** measures the frequency of inter-penetration. More details about these metrics are provided in the supplementary.

**Datasets.** We evaluate our model on NTU120-AS, Chi3D-AS and InterHuman-AS datasets with SMPL-X (Pavlakos et al., 2019) body models and actor-reactor annotations as in Petrovich et al. (2021).They contains 8118, 373 and 6022 human interaction sequences, respectively. "AS"(Xu et al., 2024) represents that they are an extended version of the original dataset (Fieraru et al., 2020; Liu et al., 2019; Trivedi et al., 2021; Liang et al., 2023), which adds annotations to distinguish actor-reactor order of each interaction sequence and SMPL-X body models for more detailed representations. We adopt the 6D rotation representation (Zhou et al., 2019) in all our experiments.

### 4.2 COMPARISON TO BASELINES

To evaluate the performance of our method, we adopt following baselines: 1) cVAE (Kingma & Welling, 2013), commonly utilized in earlier generative models for human interactions; 2) MDM (Tevet et al., 2023), the state-of-the-art diffusion-based method for human motion generation, and its variant MDM-GRU (Tevet et al., 2023), which incorporates a GRU (Cho et al., 2014) backbone; 3) AGRoL (Du et al., 2023), the current state-of-the-art method to generate full-body motions from sparse tracking signals, which adopts diffusion models with MLPs architectures; 4) ReGenNet (Xu et al., 2024), the state-of-the-art diffusion-based method for human action-reaction synthesis on online, unconstraint setting as ours. Results are taken from tables of ReGenNet (Xu et al., 2024) where all methods use 5-timestep sampling. All methods are tested on a single Nvidia 4090 GPU.

Table 1: **Comparison to state-of-the-art** on the *online, unconstrained* setting on NTU120-AS. $\rightarrow$ denotes that the result closer to Real is better, and $\pm$ represents 95% confidence interval. We highlight the best result in **Bold** and the second best in underline.

| Method | FID↓ | Acc.↑ | Div.→ | Multimod.→ | IF↓ | IV↓ |
|---|---|---|---|---|---|---|
| Real | $0.09^{\pm 0.00}$ | $0.867^{\pm 0.0002}$ | $13.06^{\pm 0.09}$ | $25.03^{\pm 0.23}$ | 21.96% | 5.35 |
| cVAE (Kingma & Welling, 2013) | $70.10^{\pm 3.42}$ | $0.724^{\pm 0.0002}$ | $11.14^{\pm 0.04}$ | $18.40^{\pm 0.26}$ | - | - |
| AGRoL (Du et al., 2023) | $44.94^{\pm 2.46}$ | $0.680^{\pm 0.0001}$ | $12.51^{\pm 0.09}$ | $19.73^{\pm 0.17}$ | - | - |
| MDM-GRU (Tevet et al., 2023) | $24.25^{\pm 1.39}$ | $0.720^{\pm 0.0002}$ | $\mathbf{13.43^{\pm 0.09}}$ | $22.24^{\pm 0.29}$ | - | - |
| MDM (Tevet et al., 2023) | $54.54^{\pm 3.94}$ | $0.704^{\pm 0.0003}$ | $11.98^{\pm 0.07}$ | $19.45^{\pm 0.20}$ | 32.63% | 17.97 |
| ReGenNet (Xu et al., 2024) | $\underline{11.00}^{\pm 0.74}$ | $\mathbf{0.749^{\pm 0.0002}}$ | $13.80^{\pm 0.16}$ | $\underline{22.90}^{\pm 0.14}$ | $\underline{13.84\%}$ | $\underline{3.50}$ |
| **ARFlow** | $\mathbf{8.07^{\pm 0.19}}$ | $\underline{0.741}^{\pm 0.0002}$ | $\underline{13.71}^{\pm 0.10}$ | $\mathbf{24.07^{\pm 0.13}}$ | **3.23**% | **0.53** |

Table 2: **Comparison to state-of-the-art** on the *online, unconstrained* setting on Chi3D-AS. $\rightarrow$ denotes that the result closer to Real is better, and $\pm$ represents 95% confidence interval. We highlight the best result in **Bold** and the second best in underline.

| Method | FID↓ | Acc.↑ | Div.→ | Multimod.→ | IF↓ | IV↓ |
|---|---|---|---|---|---|---|
| Real | $0.75^{\pm 0.18}$ | $0.691^{\pm 0.0093}$ | $7.15^{\pm 1.27}$ | $12.94^{\pm 0.96}$ | 48.80% | 33.69 |
| cVAE (Kingma & Welling, 2013) | $17.33^{\pm 17.14}$ | $0.552^{\pm 0.0024}$ | $8.20^{\pm 0.57}$ | $11.44^{\pm 0.35}$ | - | - |
| AGRoL (Du et al., 2023) | $64.83^{\pm 277.8}$ | $\underline{0.644}^{\pm 0.0039}$ | $\mathbf{7.00^{\pm 0.95}}$ | $11.33^{\pm 0.65}$ | - | - |
| MDM-GRU (Tevet et al., 2023) | $18.63^{\pm 25.87}$ | $0.574^{\pm 0.0046}$ | $6.20^{\pm 0.24}$ | $10.49^{\pm 0.32}$ | - | - |
| MDM (Tevet et al., 2023) | $18.40^{\pm 7.95}$ | $\mathbf{0.647^{\pm 0.0035}}$ | $5.89^{\pm 0.33}$ | $10.96^{\pm 0.27}$ | 58.45% | 32.64 |
| ReGenNet (Xu et al., 2024) | $\underline{13.76}^{\pm 4.78}$ | $0.601^{\pm 0.0040}$ | $6.35^{\pm 0.24}$ | $\underline{12.02}^{\pm 0.33}$ | $\underline{33.29\%}$ | $\underline{13.92}$ |
| **ARFlow** | $\mathbf{10.92^{\pm 3.70}}$ | $0.600^{\pm 0.0040}$ | $\underline{6.68}^{\pm 0.25}$ | $\mathbf{12.74^{\pm 0.17}}$ | **3.07**% | **0.03** |

**Condition-free.** For the NTU120-AS dataset in Tab. 1 and Chi3D-AS dataset in Tab. 2, our proposed ARFlow notably outperforms baselines in terms of the FID metric, demonstrating that our method better models the mapping between the action and the reaction distribution. Our method achieves the best FID and multi-modality, second best for the action recognition accuracy and diversity on the NTU120-AS dataset and the best FID and multi-modality, second best for the diversity on the Chi3D-AS dataset. For a fair comparison, we use the pre-trained action recognition model in Xu et al. (2024), so our action recognition accuracy is very close to its results. Given the restricted size of the Chi3D-AS test set, some fluctuations in the experimental results are to be expected. The results of the InterHuman-AS dataset and offline settings in Tab. B.1 and Tab. B.2 show our method also yields the best results compared to baselines. Due to our special design of generating different reaction motions for the same action, the diversity of our method is also superior to the baseline.

**Faster.** In Tab. 3, due to our condition-free design, ARFlow has a smaller number of parameters and converges faster during training, surpassing diffusion-based methods in only half training time. In the inference stage, our method also has lower latency with the same number of sampling steps.

**Bi-directional generation.** To verify the reverse generation capability of our method, we further evaluate on reaction-action tasks. As demonstrated in Tab. 4, ARFlow also completely surpasses the diffusion-based approach.

**Reprojection guidance method.** In Tab. 1 and Tab. 2, our ARFlow with RE-GUID achieves the lowest Intersection Volume, Intersection Frequency and FID than other baselines, which shows that our method achieves the lowest level of penetration while ensuring the highest generation quality. In Fig. 4, visualization results demonstrate that our method produces more physically plausible reactions. For more visualizations and **videos**, please refer to the supplementary materials.

### 4.3 ABLATION STUDY

**Network Prediction.** As depicted in Sec. 3.1, a straightforward and effective strategy is to estimate clean body poses directly through a neural network, *i.e.*, $x_1$-prediction. We compared it with $v$-prediction and the results are listed on the Prediction setting in Tab. 5 and Tab. B.3. Obviously, $x_1$-prediction has demonstrated superior performance across both settings. The reason we analyze it is that the geometric losses to regularize the generative network during the training phase directly acts on the predicted clean body poses, while $v$-prediction requires using the predicted vector field to estimate the clean poses, so the models trained by $x_1$-prediction are more effective.

**Guidance method.** As we discussed earlier, the results in Tab. 5 indicate that our reprojection guidance method (RE-GUID) completely surpasses the traditional guidance method (including higher

Table 3: Human **action-reaction** synthesis on NTU120-AS. All methods are tested on a single Nvidia 4090 GPU. **Bold** indicates the best result.

| Method | Latency(ms) | | | | Parameters(m) | Training time(h) |
|---|---|---|---|---|---|---|
| | 2-Steps | 5-Steps | 10-Steps | 100-Steps | | |
| ReGenNet | 0.33 | 0.76 | 1.58 | 15.17 | 26.80 | 48 |
| **ARFlow** | **0.05** | **0.11** | **0.23** | **2.27** | **17.87** | **24** |

Table 4: Human **reaction-action** synthesis (**reverse generation**) on NTU120-AS and Chi3D-AS.

| Method | NTU120-AS | | | | Chi3D-AS | | | |
|---|---|---|---|---|---|---|---|---|
| | FID↓ | Acc.↑ | Div.→ | Multimod.→ | FID↓ | Acc.↑ | Div.→ | Multimod.→ |
| Real | $0.01^{\pm 0.00}$ | $0.591^{\pm 0.0002}$ | $16.01^{\pm 0.10}$ | $25.78^{\pm 0.22}$ | $0.80^{\pm 0.19}$ | $0.601^{\pm 0.0090}$ | $7.17^{\pm 1.30}$ | $13.42^{\pm 0.94}$ |
| cVAE | $89.21^{\pm 4.02}$ | $0.412^{\pm 0.0004}$ | $10.01^{\pm 0.05}$ | $15.90^{\pm 0.25}$ | $46.10^{\pm 20.03}$ | $0.443^{\pm 0.0021}$ | $8.30^{\pm 0.60}$ | $9.14^{\pm 0.30}$ |
| AGRoL | $50.56^{\pm 2.61}$ | $0.390^{\pm 0.0002}$ | $11.19^{\pm 0.09}$ | $18.20^{\pm 0.16}$ | $67.90^{\pm 198.01}$ | $0.490^{\pm 0.0039}$ | $6.21^{\pm 1.10}$ | $8.89^{\pm 0.67}$ |
| MDM-GRU | $42.13^{\pm 2.20}$ | $0.424^{\pm 0.0004}$ | $12.23^{\pm 0.09}$ | $20.33^{\pm 0.30}$ | $49.03^{\pm 26.10}$ | $0.463^{\pm 0.0066}$ | $5.37^{\pm 0.25}$ | $8.09^{\pm 0.32}$ |
| MDM | $60.08^{\pm 4.15}$ | $0.403^{\pm 0.0005}$ | $10.34^{\pm 0.06}$ | $17.74^{\pm 0.21}$ | $48.20^{\pm 8.05}$ | $0.503^{\pm 0.0063}$ | $5.89^{\pm 0.39}$ | $8.41^{\pm 0.25}$ |
| ReGenNet | $36.12^{\pm 0.65}$ | $0.457^{\pm 0.0004}$ | $12.66^{\pm 0.08}$ | $19.30^{\pm 0.14}$ | $40.13^{\pm 5.30}$ | $0.484^{\pm 0.0062}$ | $5.45^{\pm 0.25}$ | $9.82^{\pm 0.32}$ |
| **ARFlow** | $\mathbf{12.81^{\pm 0.27}}$ | $\mathbf{0.486^{\pm 0.0003}}$ | $\mathbf{14.84^{\pm 0.09}}$ | $\mathbf{23.40^{\pm 0.13}}$ | $\mathbf{13.89^{\pm 3.87}}$ | $\mathbf{0.552^{\pm 0.0050}}$ | $\mathbf{6.60^{\pm 0.24}}$ | $\mathbf{12.03^{\pm 0.17}}$ |

efficiency), and it significantly reduces the damage of guidance (See Sec. D.3) to the quality of generated reaction motions. We also provide a qualitative comparison of the effects before and after using our physical constraint guidance in Fig. I.1. The qualitative and quantitative results demonstrated that our method achieves the lowest penetration level while maintaining the best quality of generated reactions.

**Number of Euler sampling timesteps.** We present comprehensive evaluation results in both online and offline scenarios, with varying Euler sampling intervals (2, 5, 10 and 100 timesteps), including the latency of reaction generation per frame on online settings and overall latency on offline settings. The experimental results, as detailed in Tab. 5 and Tab. B.3, suggest that the 5-timestep Euler sampling consistently achieves optimal performance, demonstrating superior FID scores while maintaining low latency across both evaluation settings. Thus, we adopt the 5-timestep inference as the standard configuration like Xu et al. (2024) for all the experimental results reported in this study.

Table 5: **Ablation studies** on the *online, unconstrained* setting on the NTU120-AS dataset. **Bold** indicates the best result in our method.

| Class | Settings | FID↓ | Acc.↑ | Div.→ | Multimod.→ | Latency(ms) | IF↓ | IV↓ |
|---|---|---|---|---|---|---|---|---|
| | Real | $0.085^{\pm 0.0003}$ | $0.867^{\pm 0.0002}$ | $13.063^{\pm 0.0908}$ | $25.032^{\pm 0.2332}$ | - | 21.96% | 5.35 |
| Prediction | 1) $x_1$ | $7.894^{\pm 0.1814}$ | $0.743^{\pm 0.0002}$ | $13.599^{\pm 0.1005}$ | $24.105^{\pm 0.1310}$ | - | - | - |
| | 2) $v$ | $14.726^{\pm 0.2143}$ | $0.743^{\pm 0.0002}$ | $14.154^{\pm 0.0923}$ | $23.329^{\pm 0.1125}$ | - | - | - |
| Guidance | **RE-GUID** | $\mathbf{8.073^{\pm 0.1981}}$ | $\mathbf{0.741^{\pm 0.0002}}$ | $\mathbf{13.613^{\pm 0.1004}}$ | $\mathbf{24.096^{\pm 0.1433}}$ | 0.149 | **3.23%** | **0.53** |
| | Traditional | $8.611^{\pm 0.2047}$ | $0.740^{\pm 0.0002}$ | $13.713^{\pm 0.1079}$ | $24.077^{\pm 0.1370}$ | 0.263 | 3.29% | 1.44 |
| Timesteps | 2 | $15.965^{\pm 0.2728}$ | $0.733^{\pm 0.0002}$ | $13.740^{\pm 0.0896}$ | $26.767^{\pm 0.1440}$ | 0.055 | - | - |
| | 5 | $7.894^{\pm 0.1814}$ | $0.743^{\pm 0.0002}$ | $13.599^{\pm 0.1005}$ | $24.105^{\pm 0.1310}$ | 0.111 | 8.39% | 3.26 |
| | 10 | $8.273^{\pm 0.3862}$ | $0.721^{\pm 0.0002}$ | $14.108^{\pm 0.0779}$ | $22.995^{\pm 0.1274}$ | 0.232 | - | - |
| | 100 | $8.259^{\pm 0.3902}$ | $0.747^{\pm 0.0002}$ | $14.173^{\pm 0.1024}$ | $23.619^{\pm 0.1214}$ | 2.273 | - | - |

## 5 CONCLUSION

In this work, we have presented Action-Reaction Flow Matching (ARFlow), a novel condition-free framework for human action-reaction synthesis that addresses the limitations of existing diffusion-based approaches. By establishing direct action-to-reaction mappings through flow matching, ARFlow eliminates the need for complex conditional mechanisms and supports bi-directional generation. ARFlow involves a novel reprojection guidance algorithm, RE-GUID to enable more physically plausible and efficient motion generation while preventing body penetration artifacts. Extensive evaluations on the NTU120, Chi3D and InterHuman datasets demonstrate that ARFlow excels over existing methods, showing superior performance in terms of Fréchet Inception Distance and motion diversity. Additionally, it significantly reduces body collisions, as evidenced by our introduced Intersection Volume and Intersection Frequency metrics.

**Limitations.** 1) **Method**: Although we attempt to use a reprojection method to address the issue of manifold distortions—deviations from the natural motion distribution established by flow matching, this problem still exists and the penetration loss function used may force two people to separate in some close interactions. Moreover, the generation of long-sequence reaction motions has not been explored yet. 2) **Dataset**: the dataset itself is imperfect due to inherent mocap noise. The higher IV/IF in real data show actual penetrations in captured interactions. Addressing these challenges opens a promising avenue for future research, focusing on developing advanced methods that ensure physical plausibility and motion authenticity.

## 6 Reproducibility Statement

We have elucidated our design in the paper including the model structure (Appendix. F), method parameters (Appendix. D), and the training and testing details (Sec. G). To facilitate reproduction, we will make our code and weights publicly available.

## 7 Ethics Statement

All of our experiments were conducted using publicly available and anonymized datasets. We have considered the potential social impact of our work. We acknowledge that, like any advanced technology, our method could be misused, and we strongly advise against such applications. Our work is intended for scientific advancement and positive social contributions. The authors bear full responsibility for the ethical conduct and dissemination of this research.

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

# Flow Connecting Actions and Reactions: A Condition-Free Framework for Human Action-Reaction Synthesis

## Supplementary Materials

APPENDIX

# A    ALGORITHM DERIVATION

We denote the deterministic functions: $\hat{\mathbf{x}}_1 = \mathbb{E}[\mathbf{x}_1|\mathbf{x}_t, c]$ as the $\mathbf{x}_1$-prediction, $\mathbf{v}_\theta(\mathbf{x}_t, t, c) = u_t(\mathbf{x}_t)$ as the $\mathbf{v}$-prediction. By defining the conditional probability path as a linear interpolation between $p_0$ and $p_1$, the intermediate process becomes:

$$\mathbf{x}_t = t\mathbf{x}_1 + [1 - (1 - \sigma_{\min})t]\mathbf{x}_0, \tag{18}$$

where $\sigma_{\min} > 0$ is a small amount of noise. Take the derivative of $t$ on both sides:

$$\frac{d\mathbf{x}_t}{dt} = \mathbf{x}_1 - (1 - \sigma_{\min})\mathbf{x}_0, \tag{19}$$

In the marginal velocity formula, we obtain:

$$\begin{aligned} u_t(\mathbf{x}_t) &= \mathbb{E}[\mathbf{x}_1 - (1 - \sigma_{\min})\mathbf{x}_0|\mathbf{x}_t, c] \\ &= \mathbb{E}[\mathbf{x}_1|\mathbf{x}_t, c] - (1 - \sigma_{\min})\mathbb{E}[\mathbf{x}_0|\mathbf{x}_t, c]. \end{aligned} \tag{20}$$

Substitute $\mathbf{x}_0 = \frac{\mathbf{x}_t - t\mathbf{x}_1}{1 - (1 - \sigma_{\min})t}$ from Eq. 18 into the above equation:

$$\begin{aligned} u_t(\mathbf{x}_t) &= \mathbb{E}[\mathbf{x}_1|\mathbf{x}_t, c] - (1 - \sigma_{\min})\frac{\mathbf{x}_t - t\,\mathbb{E}[\mathbf{x}_1|\mathbf{x}_t, c]}{1 - (1 - \sigma_{\min})t} \\ &= \frac{\mathbb{E}[\mathbf{x}_1|\mathbf{x}_t, c] - (1 - \sigma_{\min})\mathbf{x}_t}{1 - (1 - \sigma_{\min})t} \end{aligned}$$

where we have used the fact that $E[\mathbf{x}_t|\mathbf{x}_t] = \mathbf{x}_t$. According to $\hat{\mathbf{x}}_1 = \mathbb{E}[\mathbf{x}_1|\mathbf{x}_t, c]$ , $\mathbf{v}_\theta(\mathbf{x}_t, t, c) = u_t(\mathbf{x}_t)$, we get the equivalent form of parameterization:

$$\mathbf{v}_\theta(\mathbf{x}_t, t, c) = \frac{\hat{\mathbf{x}}_1 - (1 - \sigma_{\min})\mathbf{x}_t}{1 - (1 - \sigma_{\min})t}, \tag{21}$$

Substitute Eq. 21 into the following equation:

$$\begin{aligned} \mathbf{x}_{t'} &= \mathbf{x}_t - (t - t')\mathbf{v}_\theta(\mathbf{x}_t, t, c) \\ &= \mathbf{x}_t - (t - t')\frac{\hat{\mathbf{x}}_1 - (1 - \sigma_{\min})\mathbf{x}_t}{1 - (1 - \sigma_{\min})t} \\ &= \frac{1 - (1 - \sigma_{\min})t'}{1 - (1 - \sigma_{\min})t}\mathbf{x}_t + \frac{t' - t}{1 - (1 - \sigma_{\min})t}\hat{\mathbf{x}}_1. \end{aligned} \tag{22}$$

Finally, let $t = t_n$ and $t' = t_{n+1}$, we can obtain the estimation of $\hat{\mathbf{x}}_1$ from Eq. 21:

$$\hat{\mathbf{x}}_1 \leftarrow (1 - \sigma_{\min})\mathbf{x}_{t_n} + (1 - (1 - \sigma_{\min})t_n)\,\mathbf{v}_\theta(\mathbf{x}_{t_n}, t_n, \mathbf{c}), \tag{23}$$

and our sampling formulation based on $\mathbf{x}_1$-prediction from Eq. 22:

$$\mathbf{x}'_{t_{n+1}} \leftarrow \frac{1 - (1 - \sigma_{\min})t_{n+1}}{1 - (1 - \sigma_{\min})t_n}\mathbf{x}_{t_n} + \frac{t_{n+1} - t_n}{1 - (1 - \sigma_{\min})t_n}\,\hat{\mathbf{x}}_1. \tag{24}$$

---

**Algorithm 3** Sampling algorithm with vanilla guidance of physical constraints. ($\mathbf{v}$-prediction)

---

1: **Input**: $\mathcal{L}_{\text{pene}}$ the loss function ; $\mathbf{v}$ and $\theta$ the vector field predictor with pretrained parameters
2: **Parameters**: $N$ the number of sampling steps; $\lambda_{\text{pene}}$ the scale factor to control the strength of guidance
3: Sample $\mathbf{x}_0$ from the action distribution
4: **for** $n = 1, 2, ..., N - 1$ **do**
5:     # Estimate $\hat{\mathbf{x}}_1$ (Eq. 23)
6:     $\hat{\mathbf{x}}_1 \leftarrow (1 - \sigma_{\min})\mathbf{x}_{t_n} + (1 - (1 - \sigma_{\min})t_n)\,\mathbf{v}_\theta(\mathbf{x}_{t_n}, t_n, \mathbf{c})$
7:     # Flow maching $\mathbf{v}$-prediction sampling (Eq. 8)
8:     $\mathbf{x}'_{t_{n+1}} \leftarrow \mathbf{x}_{t_n} + (t_{n+1} - t_n)\,\mathbf{v}_\theta(\mathbf{x}_{t_n}, t_n, \mathbf{c})$
9:     # Physical constraint guidance
10:     $\mathbf{x}_{t_{n+1}} \leftarrow \mathbf{x}'_{t_{n+1}} + \lambda_{\text{pene}}\nabla_{\mathbf{x}_{t_n}}\mathcal{L}_{\text{pene}}(\hat{\mathbf{x}}_1)$
11: **end for**
12: **Return**: The reaction motion after guidance $\mathbf{x}_1 = \mathbf{x}_{t_N}$

---

# B EXTRA EXPERIMENTAL RESULTS

## B.1 INTERHUMAN-AS DATASET

For the text-conditioned setting, we adopt T2M (Guo et al., 2022), MDM (Tevet et al., 2023), MDM-GRU (Tevet et al., 2023), RAIG (Tanaka & Fujiwara, 2023) and InterGen (Liang et al., 2023) as baselines. Tab. B.1 shows our method also yields the best results compared to baselines.

Table B.1: **Comparison to state-of-the-arts** on the *online, unconstrained* setting for human action-reaction synthesis on the InterHuman-AS dataset. $\rightarrow$ denotes that the result closer to Real is better, and $\pm$ represents 95% confidence interval. We highlight the best result in **Bold**.

| Methods | R Precision (Top 3)↑ | FID↓ | MM Dist↓ | Diversity→ | MModality ↑ | IF↓ | IV↓ |
|---|---|---|---|---|---|---|---|
| Real | $0.722^{\pm0.004}$ | $0.002^{\pm0.0002}$ | $3.503^{\pm0.011}$ | $5.390^{\pm0.058}$ | - | 10.11% | 3.10 |
| T2M (Guo et al., 2022) | $0.224^{\pm0.003}$ | $32.482^{\pm0.0975}$ | $7.299^{\pm0.016}$ | $4.350^{\pm0.073}$ | $0.719^{\pm0.041}$ | - | - |
| RAIG (Tanaka & Fujiwara, 2023) | $0.363^{\pm0.008}$ | $2.915^{\pm0.0292}$ | $7.294^{\pm0.027}$ | $4.736^{\pm0.099}$ | $2.203^{\pm0.049}$ | - | - |
| InterGen (Liang et al., 2023) | $0.374^{\pm0.005}$ | $13.237^{\pm0.0352}$ | $10.929^{\pm0.026}$ | $4.376^{\pm0.042}$ | $\mathbf{2.793^{\pm0.014}}$ | - | - |
| MDM-GRU (Tevet et al., 2023) | $0.328^{\pm0.012}$ | $6.397^{\pm0.2140}$ | $8.884^{\pm0.040}$ | $4.851^{\pm0.081}$ | $2.076^{\pm0.040}$ | - | - |
| MDM (Tevet et al., 2023) | $0.370^{\pm0.006}$ | $3.397^{\pm0.0352}$ | $8.640^{\pm0.065}$ | $4.780^{\pm0.117}$ | $2.288^{\pm0.039}$ | 12.33% | 5.31 |
| ReGenNet (Xu et al., 2024) | $0.407^{\pm0.003}$ | $2.265^{\pm0.0969}$ | $6.860^{\pm0.004}$ | $5.214^{\pm0.139}$ | $2.391^{\pm0.023}$ | 9.76% | 2.93 |
| ARFlow | $\mathbf{0.434^{\pm0.003}}$ | $\mathbf{1.637^{\pm0.0413}}$ | $\mathbf{3.949^{\pm0.004}}$ | $\mathbf{5.259^{\pm0.117}}$ | $2.502^{\pm0.021}$ | **2.17%** | **0.40** |

## B.2 OFFLINE SETTINGS

To demonstrate the universality of our ARFlow, we also conducted offline setting experiments in Tab.B.2 and Tab.B.3. We replace the Transformer decoder units equipped with attention masks with an 8-layer Transformer encoder architecture just like ReGenNet (Xu et al., 2024).

Table B.2: **Results** on the **offline**, *unconstrained* setting on NTU120-AS. We highlight the best result in **Bold** and the second best in underline.

| Method | FID↓ | Acc.↑ | Div.→ | Multimod.→ | IF↓ | IV↓ |
|---|---|---|---|---|---|---|
| Real | $0.09^{\pm0.00}$ | $0.867^{\pm0.0002}$ | $13.06^{\pm0.09}$ | $25.03^{\pm0.23}$ | 21.96% | 5.35 |
| cVAE (Kingma & Welling, 2013) | $74.73^{\pm4.86}$ | $0.760^{\pm0.0002}$ | $11.14^{\pm0.04}$ | $18.40^{\pm0.26}$ | - | - |
| AGRoL (Du et al., 2023) | $16.55^{\pm1.41}$ | $0.716^{\pm0.0002}$ | $13.84^{\pm0.10}$ | $21.73^{\pm0.20}$ | - | - |
| MDM-GRU (Tevet et al., 2023) | $24.25^{\pm1.39}$ | $0.720^{\pm0.0002}$ | $\mathbf{13.43^{\pm0.09}}$ | $22.24^{\pm0.29}$ | - | - |
| MDM (Tevet et al., 2023) | $7.49^{\pm0.62}$ | $\mathbf{0.775^{\pm0.0003}}$ | $\underline{13.67^{\pm0.18}}$ | $24.14^{\pm0.29}$ | $\underline{15.45\%}$ | 3.36 |
| ReGenNet (Xu et al., 2024) | $\underline{6.19^{\pm0.33}}$ | $0.772^{\pm0.0003}$ | $14.03^{\pm0.09}$ | $\underline{25.21^{\pm0.34}}$ | 19.14% | $\underline{3.07}$ |
| **ARFlow** | $\mathbf{5.00^{\pm0.17}}$ | $\underline{0.772^{\pm0.0002}}$ | $13.84^{\pm0.09}$ | $\mathbf{25.10^{\pm0.17}}$ | **3.73%** | **0.23** |

Table B.3: **Ablation studies** on the **offline**, *unconstrained* setting on the NTU120-AS dataset. **Bold** indicates the best result in our method.

| Class | Settings | FID↓ | Acc.↑ | Div.→ | Multimod.→ | Latency(ms) |
|---|---|---|---|---|---|---|
| | Real | $0.085^{\pm0.0003}$ | $0.867^{\pm0.0002}$ | $13.063^{\pm0.0908}$ | $25.032^{\pm0.2332}$ | - |
| Prediction | 1) $x_1$ | $\mathbf{5.003^{\pm0.1654}}$ | $\mathbf{0.762^{\pm0.0002}}$ | $13.844^{\pm0.0905}$ | $\mathbf{25.104^{\pm0.1704}}$ | - |
| | 2) $v$ | $7.585^{\pm0.1562}$ | $0.757^{\pm0.0002}$ | $13.775^{\pm0.0982}$ | $24.200^{\pm0.1355}$ | - |
| Guidance | w. $\mathcal{L}_{\text{pene}}$ | $5.048^{\pm0.1167}$ | $0.750^{\pm0.0002}$ | $13.838^{\pm0.0893}$ | $25.048^{\pm0.1595}$ | - |
| Timesteps | 2 | $7.936^{\pm0.1581}$ | $0.759^{\pm0.0002}$ | $14.538^{\pm0.1016}$ | $25.904^{\pm0.1754}$ | 0.023 |
| | 5 | $\mathbf{5.003^{\pm0.1654}}$ | $\mathbf{0.762^{\pm0.0002}}$ | $13.844^{\pm0.0905}$ | $\mathbf{25.104^{\pm0.1704}}$ | 0.053 |
| | 10 | $5.506^{\pm0.1657}$ | $0.744^{\pm0.0002}$ | $13.870^{\pm0.0942}$ | $24.732^{\pm0.1533}$ | 0.110 |
| | 100 | $5.836^{\pm0.3763}$ | $0.748^{\pm0.0002}$ | $13.635^{\pm0.0948}$ | $24.058^{\pm0.1371}$ | 1.132 |
| Best | **ARFlow** | $5.003^{\pm0.1654}$ | $0.762^{\pm0.0002}$ | $13.844^{\pm0.0905}$ | $\mathbf{25.104^{\pm0.1704}}$ | 0.053 |

Table B.4: **Results** on the *online, unconstrained* setting on NTU120-AS. We highlight the best result.

| Method | FID↓ | Acc.↑ | Div.→ | Multimod.→ | Inference time per frame(ms) |
|---|---|---|---|---|---|
| Real | $0.09^{\pm0.00}$ | $0.867^{\pm0.0002}$ | $13.06^{\pm0.09}$ | $25.03^{\pm0.23}$ | - |
| cVAE (Kingma & Welling, 2013) | $70.10^{\pm3.42}$ | $0.724^{\pm0.0002}$ | $11.14^{\pm0.04}$ | $18.40^{\pm0.26}$ | - |
| AGRoL (Du et al., 2023) | $44.94^{\pm2.46}$ | $0.680^{\pm0.0001}$ | $12.51^{\pm0.09}$ | $19.73^{\pm0.17}$ | - |
| MDM (Tevet et al., 2023) | $54.54^{\pm3.94}$ | $0.704^{\pm0.0003}$ | $11.98^{\pm0.07}$ | $19.45^{\pm0.20}$ | - |
| MDM-GRU (Tevet et al., 2023) | $24.25^{\pm1.39}$ | $0.720^{\pm0.0002}$ | $\mathbf{13.43^{\pm0.09}}$ | $22.24^{\pm0.29}$ | - |
| Ready-to-React (Cen et al., 2025) | $14.02^{\pm0.75}$ | $0.729^{\pm0.0002}$ | $13.95^{\pm0.20}$ | $22.40^{\pm0.16}$ | 22 |
| ReMoS (Ghosh et al., 2024) | $24.42^{\pm2.15}$ | $0.721^{\pm0.0002}$ | $13.82^{\pm0.09}$ | $22.32^{\pm0.20}$ | - |
| ReGenNet (Xu et al., 2024) | $11.00^{\pm0.74}$ | $\mathbf{0.749^{\pm0.0002}}$ | $13.80^{\pm0.16}$ | $22.90^{\pm0.14}$ | 0.76 |
| **ARFlow** | $\mathbf{8.07^{\pm0.19}}$ | $0.741^{\pm0.0002}$ | $13.71^{\pm0.10}$ | $\mathbf{24.07^{\pm0.13}}$ | 0.11 |

## B.3 MORE RECENT BASELINES

Ready-to-React (Cen et al., 2025) incorporates a diffusion head into an auto-regressive model which is suitable for generating long sequences, but this method uses a complex multi-stage network architecture, leading to more difficult training and slower inference. To achieve real-time generation, it sacrifices somen contextual information. Due to the loss of information, its performance on our fine-grained task is slightly inferior. In contrast, our method adopts a very simple end-to-end architecture with better real-time performance and easy addition of physical constraints.

ReMoS (Ghosh et al., 2024) relies on intricate conditional mechanisms (combined spatio-temporal cross attention mechanisms) and uses a cascaded diffusion framework to generate more fine-grained reactive motions. However, it cannot achieve online generation, leading to lower performance. Tab. B.4 shows our method also yields the best results compared to baselines in both performance and efficiency.

## B.4 CONSTRAINED SETTINGS

As for constrained settings, the experimental results are as follows in Tab. B.5, where ARFlow-constrained denotes that text serves as an extra conditional input. We follow Xu et al. (2024) to process diverse prompts (from simple sentences to full sentences) using the pre-trained CLIP text encoder which has aligned the features of texts and motions. The extracted text features were fed as conditioning tokens into our model. In this setting, ARFlow-constrained achieves superior performance since the text serves as a strong hint to generate the reactions.

Table B.5: **Results** on the *online*, **constrained** setting on NTU120-AS. We highlight the best result.

| Method | FID↓ | Acc.↑ | Div.→ | Multimod.→ |
|---|---|---|---|---|
| Real | $0.09^{\pm0.00}$ | $0.867^{\pm0.0002}$ | $13.06^{\pm0.09}$ | $25.03^{\pm0.23}$ |
| ARFlow-unconstrained | $8.07^{\pm0.19}$ | $0.741^{\pm0.0002}$ | $\mathbf{13.71^{\pm0.10}}$ | $24.07^{\pm0.13}$ |
| ARFlow-constrained | $\mathbf{4.56^{\pm0.07}}$ | $\mathbf{0.893^{\pm0.0001}}$ | $12.00^{\pm0.08}$ | $\mathbf{25.06^{\pm0.21}}$ |

## B.5 APPLYING ARFLOW TO ROLE-SWITCHING INTERACTIONS

To demonstrate the bi-directional generation capability of our method to deal with interaction where the roles of actor and reactor continuously switch, we use the InterHuman-AS dataset (it has annotations to distinguish actor-reactor order and our pre-trained model can be directly reused) proposed by Xu et al. (2024) to conduct the following experiments. We select more than 150 frames long sequences where the actor-reactor orders continuously switch as the test set. Due to the training data is annotated with actor-reactor labels, we train a classifier to distinguish between actors and reactors. At the moment of inference, the classifier (Yan et al., 2018) outputs the state label of the current input motion("Actor" or "Reactor"). If it is an actor, we use ARFlow for forward inference. If the state label changes to a reactor, we use ARFlow for reverse generation.

The experimental results in Tab. B.6 show that ARFlow has significantly better performance than the diffusion-based methods that can only generate unidirectionally. Our method does not require

Table B.6: **Results** for human **role-switching** interaction synthesis on the *online, unconstrained* setting on the InterHuman-AS dataset. We highlight the best result in **Bold**.

| Methods | R Precision (Top 3)↑ | FID ↓ | MM Dist↓ | Diversity→ | MModality ↑ |
|---|---|---|---|---|---|
| Real | $0.751^{\pm0.004}$ | $0.003^{\pm0.0002}$ | $3.423^{\pm0.010}$ | $6.309^{\pm0.059}$ | - |
| MDM (Tevet et al., 2023) | $0.343^{\pm0.010}$ | $15.102^{\pm0.0962}$ | $11.550^{\pm0.095}$ | $5.201^{\pm0.140}$ | $1.983^{\pm0.046}$ |
| ReGenNet (Xu et al., 2024) | $0.364^{\pm0.008}$ | $12.190^{\pm0.1213}$ | $10.384^{\pm0.029}$ | $5.891^{\pm0.183}$ | $2.092^{\pm0.057}$ |
| **ARFlow** | $\mathbf{0.416^{\pm0.003}}$ | $\mathbf{3.886^{\pm0.0502}}$ | $\mathbf{4.260^{\pm0.009}}$ | $\mathbf{6.033^{\pm0.176}}$ | $\mathbf{2.289^{\pm0.059}}$ |

Table C.1: **Randomness Influence studies** on the *online, unconstrained* setting on the NTU120-AS dataset. **Bold** indicates the best result in our method.

| Method | Settings | FID↓ | Acc.↑ | Div.→ | Multimod.→ |
|---|---|---|---|---|---|
| | Real | $0.085^{\pm0.0003}$ | $0.867^{\pm0.0002}$ | $13.063^{\pm0.0908}$ | $25.032^{\pm0.2332}$ |
| Randomness $\beta$ | 0.05 | $13.821^{\pm0.2895}$ | $0.709^{\pm0.0003}$ | $14.002^{\pm0.1055}$ | $24.269^{\pm0.1363}$ |
| | 0.02 | $8.060^{\pm0.1517}$ | $0.728^{\pm0.0002}$ | $13.928^{\pm0.1076}$ | $24.161^{\pm0.1512}$ |
| | 0.01 | $7.671^{\pm0.1357}$ | $0.728^{\pm0.0002}$ | $13.895^{\pm0.1080}$ | $24.114^{\pm0.1486}$ |
| ARFlow | 0.001 | $7.894^{\pm0.1814}$ | $\mathbf{0.743^{\pm0.0002}}$ | $\mathbf{13.599^{\pm0.1005}}$ | $24.105^{\pm0.1310}$ |

retraining, providing a promising solution to the interaction of continuous role changes in scenarios with limited training data and resources.

# C    INFLUENCE OF SAMPLING RANDOMNESS

As depicted in Tab. C.1, although stochastic sampling increases the diversity of generated reaction motions, it sometimes has some impact on the quality of the sample due to its stochastic nature.

# D    DETAILS OF OUR GUIDANCE METHOD

## D.1    PENETRATION LOSS FUNCTON

The action-reaction task requires real-time performance. Since our network predicts joint positions, our loss function can be directly calculated with almost no additional computational overhead to meet real-time requirements. Other loss functions generally require longer computation time or introduce simulators, which is intolerable in this task. Mesh-based methods approximate mesh surface with triangular patches and then compute loss from collision triangles. Volumn-based methods require computing the occupied volume of mesh. Both of these methods also require mapping joint points to mesh surface first.

## D.2    PARAMETER ANALYSIS OF GUIDANCE STRENGTH AND WEIGHT FACTOR

We conduct a parameter analysis of guidance strength in Tab. D.1. The result of the experiments show that as the guiding strength increases, the degree of penetration between actors and reactors decreases significantly, while FID increases slightly. This is because the ground truth itself has a certain degree of penetration. Thus, this task requires our new metrics and FID to collaborate in evaluating the quality of the generated results. When the guidance strength increases to a certain extent, the decrease in penetration degree is no longer significant. Therefore, we ultimately choose $\lambda_{\text{pene}} = 2$. Our method achieves the lowest penetration level while maintaining the best generation quality. As for the weight factor, the results show that the minimum value of FID does not occur at the endpoints, thus demonstrating the effectiveness of our weighting method.

Table D.1: Parameter analysis of guidance strength and weight factor on the *online, unconstrained* setting on NTU120-AS. **Bold** indicates the best result.

| Parameter | settings | FID ↓ | IF ↓ | IV ↓ |
|---|---|---|---|---|
| | Real | 0.09 | 21.96% | 5.35 |
| $\lambda_{pene}$ | 0 | 7.89 | 8.39% | 3.26 |
| | 1 | 7.98 | 5.80% | 1.15 |
| | **2** | **8.20** | **3.54%** | **0.68** |
| | 5 | 8.49 | 1.22% | 0.13 |
| | 10 | 9.41 | 0.78% | 0.21 |
| $w$ | 0 | 8.37 | 2.71% | 0.35 |
| | 0.1 | 8.30 | 2.78% | 0.36 |
| | 0.3 | 8.19 | 2.96% | 0.37 |
| | 0.5 | 8.11 | 3.12% | 0.41 |
| | **0.7** | **8.07** | **3.23%** | **0.53** |
| | 0.9 | 8.08 | 3.32% | 0.64 |
| | 1 | 8.20 | 3.54% | 0.68 |

### D.3 LIMITATIONS OF GUIDANCE METHODS

As shown in Tab. 5, although guidance methods effectively suppress penetration, they also lead to a **slight increase** in other metrics like FID, as FID only measures the similarity between generated results and the ground truth distribution and the dataset itself are **imperfect** resulting from **inherent mocap noise**. The higher IV/IF in real data show actual penetrations in captured interactions.

### D.4 GENERALIZATION OF OUR GUIDANCE METHOD TO OTHER PHYSICAL CONSTRAINTS

We mainly propose a more accurate and efficient general guidance method(RE-GUID), in which the design of specific loss functions can be tailored to different tasks. Regarding foot sliding, we design a loss function $\mathcal{L}_{foot}$ inspired by zero velocity constraint in Zou et al. (2020) to penalize foot sliding and incorporate it into our loss function. Therefore, our overall loss function is $\mathcal{L}_{re\text{-}guid} = \mathcal{L}_{pene} + \lambda_{foot} \cdot \mathcal{L}_{foot}$, where $\lambda_{foot}$ is the foot-sliding loss weight and we set $\lambda_{foot} = 0.5$ in our experiment. For the evaluation metric of foot sliding(Skate), we follow Yuan et al. (2023) to find foot joints that contact the ground in two adjacent frames and compute their average horizontal displacement within the frames. We conduct the experiment to compare the effects before and after using $\mathcal{L}_{foot}$ in Tab. D.2. The experimental results demonstrate that our guidance method can effectively generalize to other physical constraints to generate more physically plausible motions.

Table D.2: **Generalization studies** of our guidance method (RE-GUID) to other physical constraints on the *online, unconstrained* setting on NTU120-AS. We highlight the best result in **Bold**.

| Method | FID↓ | Acc.↑ | Div.→ | Multimod.→ | IF↓ | IV↓ | Skate↓ |
|---|---|---|---|---|---|---|---|
| Real | $0.09^{\pm 0.00}$ | $0.867^{\pm 0.0002}$ | $13.06^{\pm 0.09}$ | $25.03^{\pm 0.23}$ | 21.96% | 5.35 | 0.65 |
| ARFlow | $8.07^{\pm 0.19}$ | $0.741^{\pm 0.0002}$ | $\mathbf{13.71^{\pm 0.10}}$ | $24.07^{\pm 0.13}$ | 3.23% | **0.53** | 1.74 |
| ARFlow w. $\mathcal{L}_{foot}$ | $\mathbf{7.90^{\pm 0.16}}$ | $\mathbf{0.743^{\pm 0.0002}}$ | $13.73^{\pm 0.11}$ | $\mathbf{24.17^{\pm 0.12}}$ | 3.23% | 0.55 | **0.70** |

## E MORE RELATED WORK

**Human Motion Generation.** Human motion synthesis aims to generate diverse and realistic human-like motion conditioned on different guidances (Zhang et al., 2023b; Zhou & Wang, 2023; Ao et al., 2022). Recently, many diffusion-based motion generation models have been proposed (Zhang et al., 2022; Chen et al., 2023; Wu et al., 2024) and demonstrate better quality compared to alternative models such as VAE (Guo et al., 2020; Cervantes et al., 2022), flow-based models (Rezende & Mohamed, 2015; Aliakbarian et al., 2022) or GANs (Yan et al., 2019; Xu et al., 2022). Alternatively, motion can be regarded as a new form of language and embedded into the language model

framework (Zhang et al., 2023a; Jiang et al., 2023). Meanwhile, the exploration of guiding the sampling process of diffusion models (Chung et al., 2023; Yang et al., 2024) has been a key area in motion diffusion models, PhysDiff (Yuan et al., 2023) proposes a physics-guided motion diffusion model, which incorporates physical constraints in a physics simulator into the diffusion process. GMD (Karunratanakul et al., 2023) presents methods to enable spatial guidance without retraining the model for a new task. DNO (Karunratanakul et al., 2024) proposes a motion editing and control approach by optimizing the diffusion latent noise of an existing pre-trained model.

## F  DETAILS OF OUR FRAMEWORK

We present our Human Action-Reaction Flow Matching (ARFlow) framework, illustrated in Fig. 2, which comprises a flow module and a Transformer decoder $G$. Given a paired action-reaction sequence and an optional signal $c$ (e.g. , an action label, dotted lines in Fig. 2), $<x_0^{1:H}, x_1^{1:H}, c>$, $x_1^{1:H}$ represents the reaction to generate. For a sampled timestep $t$, we linearly interpolate $x_1^{1:H}$ and $x_0^{1:H}$ as Eq. 4 to produce the $x_t^{1:H}$. Then the $x_t^{1:H}$ turns into the latent features through an FC layer to dimension $d$. The timestep $t$ and the optional condition $c$ are separately projected to dimension $d$ using feed-forward networks and combined to form the token $z$. The Transformer decoder $G$, implemented with stacked 8 layers, prevents future information leakage through masked multi-head attention, enabling *online* generation as in Xu et al. (2024). Decoder $G$ takes $z$ as input tokens and $x_t^{1:H}$ combined with a standard positional embedding as output tokens, along with a directional attention mask to ensure the model cannot access future actions at the current timestep. The decoder's output is projected back to produce the predicted clean body poses $\hat{x}_1^{1:H}$. Online reaction generation is achieved in an auto-regressive manner, following the approach of Xu et al. (2024). The intention branch can be activated when the actor's intention is accessible to the reactor, or deactivated otherwise. The directional attention mask can be turned off for offline settings.

At the inference stage, we employ our physical constraint guidance method. After training latent linear layers and Transformer decoder $G$, our ARFlow uses them for $x_1$-prediction based sampling. The sampling process is further guided by the gradient of $\mathcal{L}_{pene}$ to generate physically plausible reactions.

## G  IMPLEMENTATION DETAILS

Our ARFlow model is trained with $T = 1,000$ timesteps using a classifier-free approach (Ho & Salimans, 2021). The number of decoder layers is 8 and the latent dimension of the Transformer tokens is 512. The batch size is configured as 32 for NTU120-AS, InterHuman-AS and 16 for Chi3D-AS. The interaction loss weight is set to $\lambda_{\text{inter}} = 1$. Each model is trained for 500K steps on single NVIDIA 4090 GPU within 48 hours. During inference, unless otherwise stated, we employ 5-timestep sampling for all the diffusion-based and our models in our experiments as Xu et al. (2024) for a fair comparison. For the physical constraint guidance, we set the safe distance $\zeta = 0.5$, $\lambda_{\text{pene}} = 2$ and $w = 0.7$ .

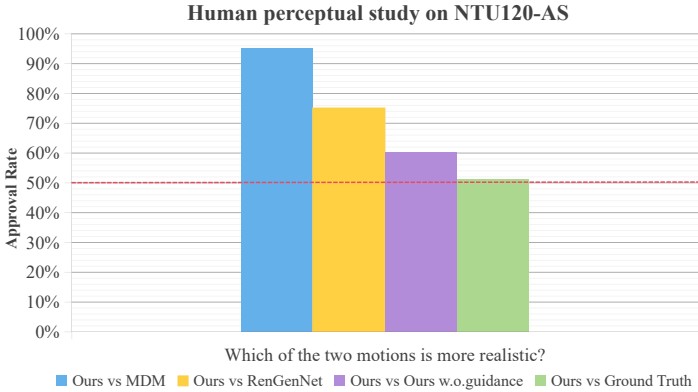

Figure G.1: Human perceptual study results on NTU120-AS.

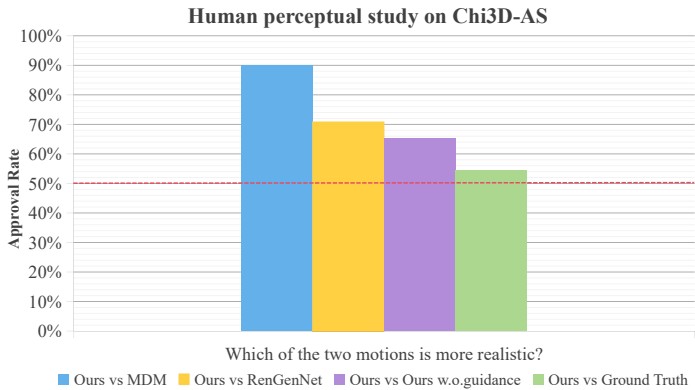

Figure G.2: Human perceptual study results on Chi3D-AS.

## H DETAILS OF THE METRIC CALCULATIONS

We follow the prior works in human action-reaction synthesis, ReGenNet (Xu et al., 2024) and MDM (Tevet et al., 2023) to calculate the Frechet Inception Distance(FID) (Heusel et al., 2017), action recognition accuracy, diversity and multi-modality. For a fair comparison, we use the pre-trained action recognition model in Xu et al. (2024), which is a slightly modified version of ST-GCN (Yan et al., 2018). The model takes the 6D rotation representation of the SMPL-X parameters as input and outputs classification results of action-reaction pairs. We generate 1,000 reaction samples by sampling actor motions from test sets and evaluate each method 20 times using different random seeds to calculate the average with the 95% confidence interval.

1) Frechet Inception Distance (FID) (Heusel et al., 2017) measures the similarity in feature space between predicted and ground-truth motion; 2) Action Recognition Accuracy (Acc.) assesses how likely a generated motion can be successfully recognized. We adopt the pre-trained ST-GCN model to classify the generated results; 3) Diversity (Div.) evaluates feature diversity within generated motions. Given the motion feature vectors of generated motions and real motions as $\{v_1, \cdots, v_{S_d}\}$ and $\{v'_1, \cdots, v'_{S_d}\}$, the diversity is defined as $Diversity = \frac{1}{S_d} \sum_{i=1}^{S_d} ||v_i - v'_i||_2$. $S_d = 200$ in our experiments. 4) Multi-modality (Multimod.) quantifies the ability to generate multiple different motions for the same action type. Given a collection of motions containing $C$ action types, for $c$-th action, we randomly sample two subsets of size $S_l$, and then extract the corresponding feature vectors as $\{v_{c,1}, \cdots, v_{c,S_l}\}$ and $\{v'_{c,1}, \cdots, v'_{c,S_l}\}$, the multimodality is defined as $Multimod. = \frac{1}{C \times S_l} \sum_{c=1}^{C} \sum_{i=1}^{S_l} ||v_{c,i} - v'_{c,i}||_2$. $S_l = 20$ in our experiments.

**Physical Metrics.** To qualitatively measure the degree of penetration, we introduced two metrics:

1) **Intersection Volume (IV).** Penetrate in Yuan et al. (2023); Han et al. (2024) just measures ground penetration which is not suitable for measuring the degree of penetration between humans. Interpenetration in Liu et al. (2024) can only be computed as rigid bodies in the physics simulation. Inspired by Solid Intersection Volume (IV) (Zhou et al., 2022; Liu & Yi, 2024), we measure human-human inter-penetration by reporting the volume occupied by two human meshes, *i.e.*

$$IV = \frac{1}{H \cdot N_{\text{total}}} \sum_{i=1}^{N_{\text{total}}} \sum_{h=1}^{H} V_{\text{pene}}^h, \qquad (25)$$

where $V_{\text{pene}}^h$ represents intersection volume of frame $h$ and $N_{\text{total}}$ denotes the total number of samples.

2) **Intersection Frequency (IF).** Inspired by Contact Frequency in Li et al. (2024); Siyao et al. (2024), we introduce IF to measure the frequency of inter-penetration, *i.e.*

$$IF = f_{\text{pene}}/F_{\text{total}}, \qquad (26)$$

where $f_{\text{pene}}$ represents the number of inter-penetration frames and $F_{\text{total}}$ is the total number of frames. We generate 260 samples for evaluation.

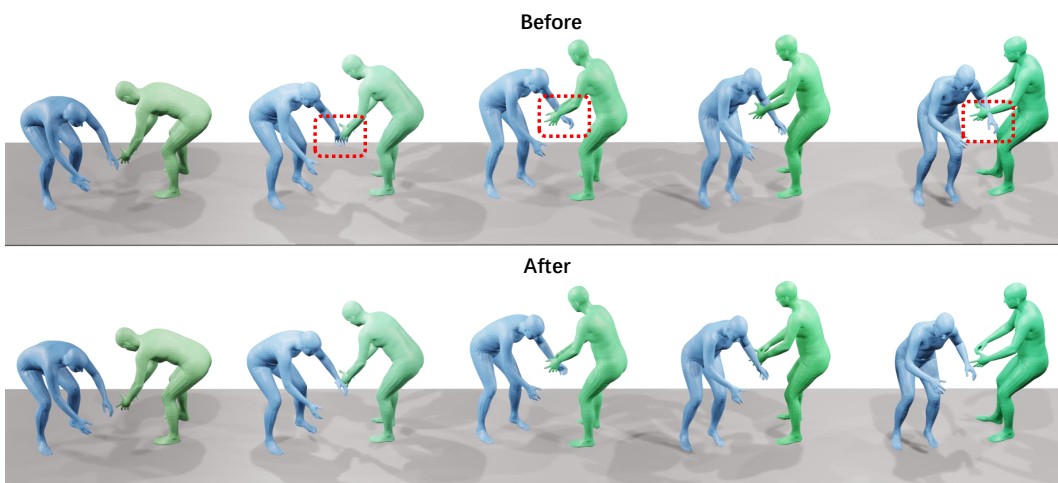

Figure I.1: **Visualization comparison** of the effects before and after using physical constraint guidance. Blue for actors and Green for reactors.

## I  USER STUDY

We conducted a human perceptual study to investigate the quality of the motions generated by our model. We invite 20 users to provide four comparisons. For each comparison, we ask the users "Which of the two motions is more realistic?", and each user is provided 10 sequences to evaluate.

The results are shown in Fig. G.1 and Fig. G.2. Our results were preferred over the other state-of-the-art and are even competitive with ground truth motions.

## J  EXTRA QUALITATIVE RESULTS

We show the generated motions of our method against others in Fig. J.1. We highlight the implausible motions in rectangle marks, it is clear that our method learns the correct reactions and avoids human-human inter-penetrations as much as possible.

**Failure case.** We also show the failure cases of our motion generation pipeline in Fig. J.2. Our current method's constraint on physical plausibility only relies on a penetration loss function to guide the sampling process, which may not be sufficient to capture intricate fine-grained interactions that require people contact but do not allow penetration. For example, in the case of handshaking, when two people's hands penetrate, our guidance method will force their hands to separate to prevent penetration. In future work, we can refer to some fine-grained loss functions in hand-object interaction (Tian et al., 2024; Lee et al., 2024) to ensure that two people's hands contact but do not penetrate. For some other physical plausibility issues, such as foot-skating, we can also incorporate these corresponding physical constraints (Zou et al., 2020; Yuan et al., 2023) into our RE-GUID method to address these issues.

## K  BROADER IMPACTS

Our model demonstrates significant potential for AR/VR and gaming applications by enabling the generation of plausible human reactions. Beyond virtual environments, the proposed approach provides an innovative technical pathway for real-world human-robot interaction, where motion patterns can be transferred to robotic systems through motion remapping technology. Although this advancement may inspire future research, we acknowledge potential misuse risks similar to other generative models, warranting ethical considerations as the technology develops.

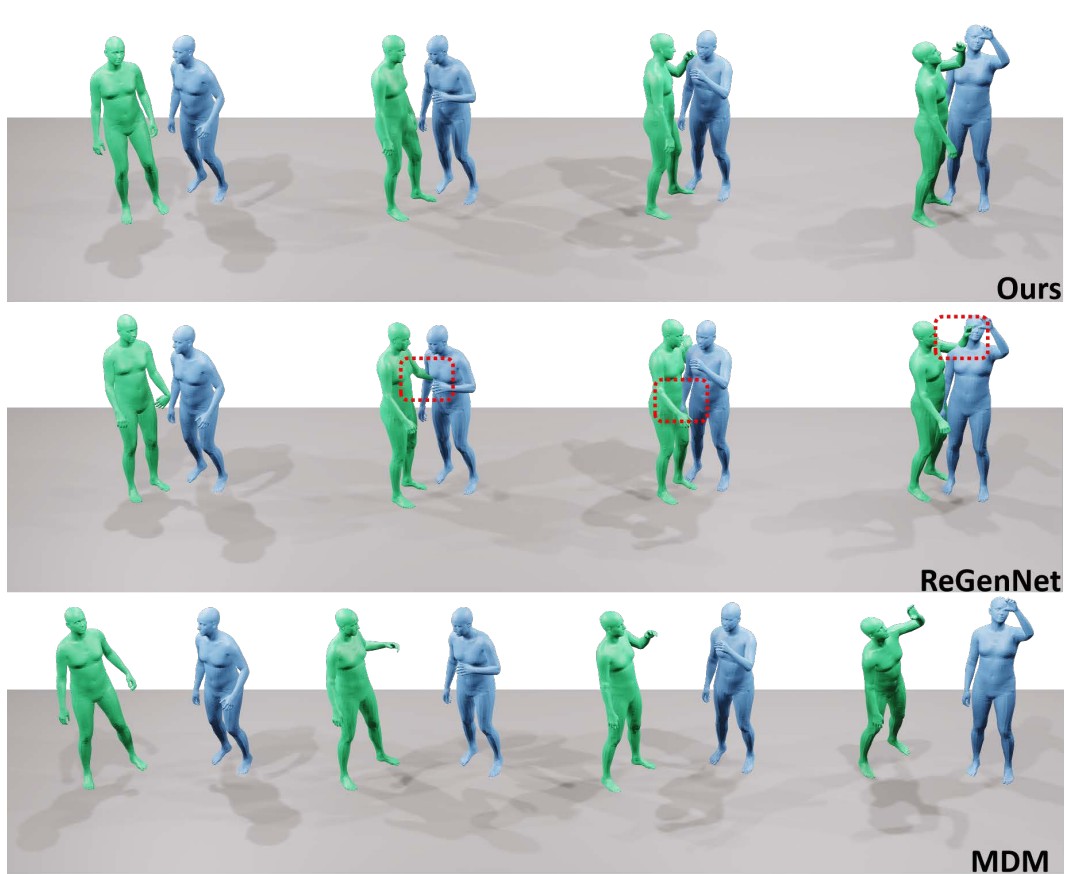

Figure J.1: The extra qualitative experiment. Blue for actors and Green for reactors.

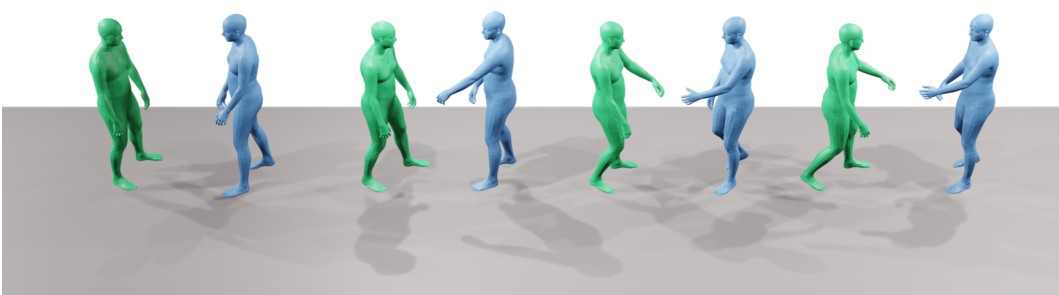

Figure J.2: Failure case of our method.

## L    USE OF LLMs

During the preparation of this work, we only use large language models to check grammar, proofread and improve linguistic fluency. All suggestions provided by the LLM have been thoroughly reviewed, validated, and integrated by the authors. The authors take full responsibility for the originality and integrity of the content presented in this paper.

