# OpenReview forum: "Flow Connecting Actions and Reactions: A Condition-Free Framework for Human Action-Reaction Synthesis"
_ICLR.cc/2026/Conference — Submitted to ICLR 2026_

### Official Review · Reviewer_6px3 · 2025-10-28

**Soundness:** 2
**Presentation:** 3
**Contribution:** 3
**Rating:** 4
**Confidence:** 4

**Summary:**

The paper present a new method for human reaction motion generation by using flow matching adapted for action to reaction synthesis. Additionaly, a guidance module is build to correct error that can appear from using the flow matching, correcting direction and body penetration. Experiments are performed on 3 commonly used datasets and higly the stenghts of the methods especially with regard to avoiding physical impossibilities.

**Strengths:**

- The use of flow matching for reaction generation is interesting to avoid issue encountered by previous method while being  faster than diffusion based methods.
- The proposed guidance module seems to improve performance while being faster than traditional guidance.
- The proposed methods outperforms the other approachs on the three datasets proposed in the study quantitatively and qualitatively.
- Methods and experiments are very detailed and well explained
- The new metrics to check for body penetration are interesting and should be of interest to the community

**Weaknesses:**

1. Some more recent are missing for the comparison [1,2], while I am not sure if they are useable in the online setting proposed by the authors it would have been interesting to have the comparison at least for the offline setting.
2. There are two claims about the ability of the methods to deal with challenges but they are never investigated in the experiments. i.e. boxing with recursively generating the motion of both humans ([1] propose a boxing dataset), the condition signal (action motion) is stated to be replaceable by text or audio. Showing results in both scenari would really strengthen the paper.
3. The three datasets contain only relatively simple and short interactions. It would have been interesting to see results on more complex and longer motions.
4. The IF and IV values are much lower for the proposed method than for the GT but this is never explained
5. Table B1 and B2 are missing the IV and IF values
6. It would have been interesting to see the results of reverse generation for all method to highlight their limitations.


[1] Ready-to-React: Online Reaction Policy for Two-Character Interaction Generation, ICLR 2025
[2] REMOS: 3D Motion-Conditioned Reaction Synthesis for Two-Person Interactions, ECCV 2024

**Questions:**

questions:
- see weaknesses. Particularly answers to 1 and 2 could make me increase my rating

suggestions:
- Interhuman is mentionned several times in the paper but the results are only in the appendix, this should be made more clear before referencing to table B1.
- Table 3 show the methods speed but the gpu used is only mentioned in the appendix. It should be mentionned in the main paper instead.

---

> ### Author Response · Authors · 2025-11-27
> **Official Comment by Authors (1/2)**
>
> Thank you for the detailed review and thoughtful feedback. Please find our responses to your questions below.
>
> >Q1:Some more recent are missing for the comparison [1,2], while I am not sure if they are useable in the online setting proposed by the authors it would have been interesting to have the comparison at least for the offline setting.
>
> **A1:**  To compare the capability of these methods in our online setting for its practical significance, we have supplemented the experiments as follows
>
> | Method         | FID↓  | Acc.↑ | Div.→ | MM.→  | Inference time per frame (ms)↓ |
> | -------------- | ----- | ----- | ----- | ----- | :-----: |
> | Real           | 0.09  | 0.87  | 13.06 | 25.03 | -        |
> | Ready-to-React | 14.02 | 0.73  | 13.95 | 22.40 | 22        |
> | ReMoS          | 24.42 | 0.72  | 13.82 | 22.32 | ∞          |
> | ReGenNet       | 11.00 | **0.75**  | 13.80 | 22.90 | 0.76      |
> | **ARFlow**         | **8.07**  | 0.74  | **13.71** | **24.07** | **0.11**     |
>
> **Ready-to-React** incorporates a diffusion head into an auto-regressive model which is suitable for generating long sequences, but this method uses a complex multi-stage network architecture, leading to more difficult training and slower inference. To achieve real-time generation, it sacrifices some contextual information. Due to the loss of information, its performance on our fine-grained task is inferior. In contrast, our method adopts a very **simple** end-to-end architecture with better **real-time** performance and easy addition of physical constraints.
>
> **ReMoS** relies on intricate conditional mechanisms (combined spatio-temporal cross attention mechanisms) and uses a cascaded diffusion framework to generate more fine-grained reactive motions. However, it cannot achieve online generation, leading to lower performance. We have added the above analysis and experimental results in **Appendix B.3** marked with blue color.
>
> >Q2: There are two claims about the ability of the methods to deal with challenges but they are never investigated in the experiments. i.e. boxing with recursively generating the motion of both humans ([1] propose a boxing dataset), the condition signal (action motion) is stated to be replaceable by text or audio. Showing results in both scenario would really strengthen the paper.
>
> **A2:**
> (1) To demonstrate the bi-directional generation ability of our method to deal with interactions where the roles of actor and reactor continuously **switch**, we use the InterHuman-AS dataset (it has annotations to distinguish actor-reactor order and our pre-trained model can be directly reused) proposed by ReGenNet [1] to conducted the following experiments. (The boxing dataset currently lacks annotations and partitioning, and is only limited to boxing tasks.) We select more than 150 frames long sequences where the actor-reactor orders continuously switch as the test set. Due to the training data is annotated with actor-reactor labels, we train a **classifier** to distinguish between actors and reactors. At the moment of inference, the classifier outputs the **state label** of the current input motion("Actor" or "Reactor"). If it is an actor, we use ARFlow for **forward** inference. If the state label changes to a reactor, we use ARFlow for **reverse** generation.
>
> | Method   | R Precision↑ | FID↓  | MM Dist.↓ | Div.→ | MM.→ |
> | :---: | :---: | :---: | :---: | :---: | :---: |
> | Real     | 0.75         | 0.01  | 3.42      | 6.31  | -    |
> | MDM      | 0.34         | 15.10 | 11.55     | 5.20  | 1.98 |
> | ReGenNet | 0.36         | 12.19 | 10.38     | 5.89  | 2.09 |
> | **ARFlow**   | **0.42**         | **3.89**  | **4.26**  | **6.03**  | **2.29** |
>
> The experimental results show that ARFlow has significantly better performance than the diffusion-based methods that can only generate unidirectionally. Our method **does not require retraining**, providing a promising solution to the interaction of continuous role changes in scenarios with **limited** training data and resources.
>
> (2) As for condition signals, the experimental results are as follows, where ARFlow-constrained denotes that condition signal is replaced by text and this setting achieves superior performance since the text serves as a strong hint to generate the reactions.
>
> | Method               | FID↓ | Acc.↑ | Div.→ | MM.→  |
> | -------------------- | ---- | ----- | ----- | ----- |
> | Real                 | 0.09 | 0.87  | 13.06 | 25.03 |
> | ARFlow-unconstrained | 8.07 | 0.74  | 13.71 | 24.07 |
> | ARFlow-constrained   | 4.56 | 0.89  | 12.00 | 25.06 |
>
> We hope this explanation can address the reviewer’s concern and provide clarity on our claims. We have added the above analysis and experimental results in **Appendix B.5** and **Appendix B.4** marked with blue color.
>
> **Reference:**
>
> [1] Liang Xu, et al. ReGenNet: Towards Human Action-Reaction Synthesis. CVPR, 2024.

---

> > ### Comment · Reviewer_6px3 · 2025-11-27
> >
> > I thank the authors for their answers. I still have a few questions regarding answer A2.
> >
> > 1. Could the authors share more details regarding the text conditionned experiment? Here I don't really understand how the experiment is conducted ; i.e. what do the text condition contains (single words, simple sentences,long sentences, action verbs...) and how it is processed (e.g. CLIP encoding...)....
> > 2. When saying that "The boxing dataset currently lacks annotations and partitioning" does that mean that the proposed apporach needs specific actor/reactor roles? The method being natively able to do both forward and reverse generation, I would have expected it not to be reliant on clearly defined role or motion type.
> > 3. It would have been interesting to have qualitative results, especially videos, for the role switching experiment.

---

> > > ### Author Response · Authors · 2025-12-03
> > >
> > > We sincerely thank the reviewer for the prompt response.
> > >
> > > >Q1: Could the authors share more details regarding the text conditionned experiment? Here I don't really understand how the experiment is conducted ; i.e. what do the text condition contains (single words, simple sentences,long sentences, action verbs...) and how it is processed (e.g. CLIP encoding...)....
> > >
> > > **A1:** Thank you. For our text-conditioned experiments, we follow [1] to process diverse prompts (from simple sentences to full sentences) using the pre-trained CLIP text encoder which has aligned the features of texts and motions. The extracted text features were fed as conditioning tokens into our flow matching model. We will add these details to the paper.
> > >
> > > >Q2: When saying that "The boxing dataset currently lacks annotations and partitioning" does that mean that the proposed apporach needs specific actor/reactor roles? The method being natively able to do both forward and reverse generation, I would have expected it not to be reliant on clearly defined role or motion type.
> > >
> > > **A2:** Thank you for raising this point. Our method is **not** reliant on pre-defined role or motion type due to our bi-directional generation design. The need for role annotation here is specific to structuring the asymmetric action-reaction task for evaluation. We actually want to highlight that our action-reaction model can be **directly reused** without retraining, providing a promising solution to the role-switching interactions in scenarios with **limited** training data and resources. Because the current high-quality and diverse human-human/robot interaction dataset is not particularly sufficient.
> > >
> > > >Q3: It would have been interesting to have qualitative results, especially videos, for the role switching experiment.
> > >
> > > **A3:** Thank you for the suggestion. As requested, we have included new qualitative results in the supplementary video (**4:00-4:30**), showcasing **role-switching** examples in boxing and fencing. These visualizations effectively demonstrate our method's unique bi-directional generation capability, allowing the same model to synthesize both "actor" and "reactor" motions. This evidence further strengthens our paper's central claim regarding flexible interaction modeling.
> > >
> > > **Reference:**
> > >
> > > [1] Liang Xu, et al. ReGenNet: Towards Human Action-Reaction Synthesis. CVPR, 2024.

---

> ### Author Response · Authors · 2025-11-27
> **Official Comment by Authors (2/2)**
>
> >Q3: The three datasets contain only relatively simple and short interactions. It would have been interesting to see results on more complex and longer motions.
>
> **A3:** Long and more complex sequence generation is also a limitation of diffusion-based methods [1] and ours (See [1] Limitations and ours Line. 489-490). Our work mainly explores a new modeling paradigm in this task. To generate long and more complex sequences in future work, we can add a fixed length historical reaction motion sequence as model input or draw inspiration from [2]. We can combine the long sequence generation ability of autoregressive models with the modeling ability of diffusion models for complex sequences to generate long and more complex motions.
>
> >Q4: The IF and IV values are much lower for the proposed method than for the GT but this is never explained
>
> **A4:** We explained this phenomenon in Appendix D.3 (Line. 1044-1045). The higher IV/IF in real data stems from **inherent mocap noise** causing actual penetrations in captured interactions (ReGenNet [1] also explicitly notes it in their limitations). Due to the introduction of physical constraints to prevent penetration, the IF and IV values in our method are much lower. In the revision, we have emphasized this phenomenon again in the Limitations section for greater clarity.
>
> >Q5: Table B1 and B2 are missing the IV and IF values
>
> **A5:** We appreciate your feedback and the missing IV and IF values (we have added them in the revision) for Table B1 and B2 are as follows
>
> | Method   | FID↓ |  IF↓ | IV↓  |
> | ----- | :---: | :---: | :---: |
> | Real     | 0.01 | 10.11% | 3.10 |
> | MDM      | 3.40 | 12.33% | 5.31 |
> | ReGenNet | 2.27 | 9.76%  | 2.93 |
> | **ARFlow**   | **1.64** | **2.17%**  | **0.40** |
>
> | Method   | FID↓ |  IF↓ | IV↓  |
> | -------- | :---: | :---: | :---: |
> | Real     | 0.09 | 21.96% | 5.35 |
> | MDM      | 7.49 | 15.45% | 3.36 |
> | ReGenNet | 6.19 | 19.14% | 3.07 |
> | **ARFlow**   | **5.00** | **3.73%**  | **0.23** |
>
> >Q6: It would have been interesting to see the results of reverse generation for all method to highlight their limitations.
>
> **A6:** Thanks for your feedback. We have supplemented the experiments of reverse generation for all methods in our revision and the experimental results further highlight the reverse generation capability of our method.
>
> | Method   | FID↓  | Acc.↑ | Div.→ | MM.→  |
> | -------- | ----- | ----- | ----- | ----- |
> | Real     | 0.01  | 0.59  | 16.01 | 25.78 |
> | cVAE     | 89.21 | 0.41  | 10.01 | 15.90 |
> | AGRoL    | 50.56 | 0.39  | 11.19 | 18.20 |
> | MDM-GRU  | 42.13 | 0.42  | 12.23 | 20.33 |
> | MDM      | 60.08 | 0.40  | 10.34 | 17.74 |
> | ReGenNet | 36.12 | 0.46  | 12.66 | 19.30 |
> | **ARFlow**   | **12.81** | **0.49**  | **14.84** | **23.40** |
>
> > Suggestion1: Interhuman is mentioned several times in the paper but the results are only in the appendix, this should be made more clear before referencing to table B1.
>
> > Suggestion2: Table 3 show the methods speed but the gpu used is only mentioned in the appendix. It should be mentioned in the main paper instead.
>
> **A7:** We have followed your suggestions and emphasized interhuman in bold in the main paper before referencing to table B1, and illustrated the use of GPU in the main paper and Table 3.
>
> We sincerely hope the efforts we made can address your concerns.
>
> **Reference:**
>
> [1] Liang Xu, et al. ReGenNet: Towards Human Action-Reaction Synthesis. CVPR, 2024.
>
> [2] Boyuan Chen, et al. Diffusion Forcing: Next-token Prediction Meets Full-Sequence Diffusion. NeurIPS, 2024.

---

### Official Review · Reviewer_fcJX · 2025-10-31

**Soundness:** 3
**Presentation:** 3
**Contribution:** 2
**Rating:** 6
**Confidence:** 4

**Summary:**

This paper proposes ARFlow, a human action-reaction synthesis framework that establishes direct mappings between action and reaction distributions through flow matching. ARFlow eliminates complex conditional mechanisms in traditional diffusion models and enables bidirectional motion generation while maintaining real-time capability. The framework incorporates a reprojection guidance method  that corrects initial point deviation during sampling and significantly reduces bodily inter-penetration between characters. Extensive experiments demonstrate that ARFlow achieves superior performance in motion quality, diversity and physical plausibility, outperforming existing methods in both online and unconstrained settings.

**Strengths:**

1. The proposal presents a novel application of flow matching to action-reaction synthesis, establishing direct mappings between action and reaction distributions. This eliminates the need for complex conditional mechanisms in diffusion-based methods and enables bidirectional generation.

2. The paper accurately identifies the Initial Point Deviation issue in flow matching sampling and designs the RE-GUID reprojection guidance method. This method corrects the deviation without requiring differentiation of the neural network, balancing physical plausibility and motion generation quality while effectively reducing body penetration.

3. Through extensive experiments, the proposed method is shown to outperform existing baselines on common motion metrics and drastically reduce body collisions, while also achieving simpler training and faster inference.

**Weaknesses:**

1. The penetration loss function might force characters to separate in close interactions, failing to balance physical plausibility and natural interaction dynamics.

2. Although the paper visualizes a failure case in Figure J.2, it lacks a systematic analysis of failure patterns, weakening the completeness of the method’s limitation discussion

**Questions:**

1. Are there specific motion categories or interaction patterns where the model exhibits significant performance degradation?

2. Regarding the physical constraint guidance, could its formulation be extended to address a wider range of physical plausibility aspects, such as foot-skating? Furthermore, I am interested in seeing ARFlow's performance evaluated on more physical plausibility metrics to better understand its capabilities and limitations.

3. Would the physical constraint guidance struggle to maintain consistent physical plausibility across long sequences, resulting in occasional penetration or unnatural motions in later frames?

---

> ### Author Response · Authors · 2025-11-27
>
> Thank you for the detailed review and thoughtful feedback. Please find our responses to your questions below.
>
> >Q1: The penetration loss function might force characters to separate in close interactions, failing to balance physical plausibility and natural interaction dynamics.
>
> **A1:** Yes, we claimed this issue in our Limitations (Line. 489). But we can adjust the hyperparameters $\lambda_{pene}$ in Eq.(13) that controls the strength of the guidance to balance physical plausibility and natural interaction dynamics. In future work, integrating more fine-grained loss functions to our guidance method may address this issue.
>
> >Q2: Although the paper visualizes a failure case in Figure J.2, it lacks a systematic analysis of failure patterns, weakening the completeness of the method’s limitation discussion.
>
> **A2:** Thanks for your thoughtful feedback. Our current method's constraint on physical plausibility only relies on a penetration loss function to guide the sampling process, which may not be sufficient to capture intricate fine-grained interactions that require people contact but do not allow penetration. For example, in the case of handshaking, when two people's hands penetrate, our guidance method will force their hands to separate to prevent penetration. In future work, we can refer to some fine-grained loss functions in hand-object interaction[1,2] to ensure that two people's hands contact but do not penetrate. We have supplemented the systematic analysis in the failure case section (Line.1217 -1224) marked with blue color in Appendix J.
>
> >Q3: Are there specific motion categories or interaction patterns where the model exhibits significant performance degradation?
>
> **A3:** Yes, as mentioned in the previous question, our method is not yet effective in some fine-grained interactions which require precise contact. In future work, we can add more fine-grained loss functions to our guidance method to address these issues.
>
> >Q4: Regarding the physical constraint guidance, could its formulation be extended to address a wider range of physical plausibility aspects, such as foot-skating? Furthermore, I am interested in seeing ARFlow's performance evaluated on more physical plausibility metrics to better understand its capabilities and limitations.
>
> **A4:** Yes. Our RE-GUID is a general guidance method, in which the design of specific loss functions can be tailored to different tasks. Regarding foot-skating, we refer to [3] and designed a loss function $L_{foot}$ to penalize foot-skating, which is incorporated into our loss function. We follow [4] to compute the evaluation metric of foot-skating(Skate). The following experimental results compare the effects before and after using this physical constraint. More details are referred to **Appendix D.4** marked with blue color please.
>
> | Method          | FID↓ | Acc.↑ | Div.→ | MM.→  | IF↓    | IV↓  | Skate↓ |
> | --------------- | ---- | ----- | ----- | ----- | ------ | ---- | ------ |
> | Real            | 0.09 | 0.87  | 13.06 | 25.03 | 21.96% | 5.35 | 0.65   |
> | ARFlow          | 8.07 | 0.74  | **13.71** | 24.07 | 3.23%  | **0.53** | 1.74   |
> | ARFlow w. $L_{foot}$ | **7.90** | 0.74  | 13.73 | **24.17** | 3.23%  | 0.55 | **0.70**   |
>
> >Q5: Would the physical constraint guidance struggle to maintain consistent physical plausibility across long sequences, resulting in occasional penetration or unnatural motions in later frames?
>
> **A5:** Thanks for your valuable feedback. Indeed, as the sequence becomes longer, the weight of the later generated sequence in the loss function will become smaller and smaller. However, due to the design of our efficient guidance method, we can optimize the generation process of each frame. For very long sequences, we can choose to use a sliding window mechanism to calculate our loss only for the sequence within the window to ensure the accuracy of the gradient.
>
>
> **Reference:**
>
> [1] Jie Tian, et al. Gaze-guided Hand-Object Interaction Synthesis: Dataset and Method. arXiv e-prints, pp. arXiv–2403, 2024.
>
> [2] Jihyun Lee, et al. InterHandGen: Two-Hand Interaction Generation via Cascaded Reverse Diffusion. CVPR, 2024.
>
> [3] Yuliang Zou, et al. Reducing Footskate in Human Motion Reconstruction with Ground Contact Constraints. CVPR, 2020.
>
> [4] Ye Yuan, et al. Physdiff: Physics-guided Human Motion Diffusion Model. ICCV, 2023.

---

> ### Comment · Reviewer_fcJX · 2025-11-28
>
> The authors have addressed my questions. I will keep my rating.

---

### Official Review · Reviewer_EMJ3 · 2025-11-01

**Soundness:** 2
**Presentation:** 2
**Contribution:** 2
**Rating:** 4
**Confidence:** 3

**Summary:**

The paper proposes ARFlow, a flow-matching framework for human action-reaction synthesis, claiming to replace noise-conditioned diffusion models with a direct mapping between action and reaction motions. It introduces a reprojection guidance (RE-GUID) method to reduce body penetration. Experiments on NTU120-AS, Chi3D-AS, and InterHuman-AS report lower FID and fewer penetrations compared with diffusion baselines.

**Strengths:**

Flow matching is an interesting alternative to diffusion and has potential for faster inference. This paper introduces Flow matching for reaction synthesis for two person interaction

The proposed RE-GUID physically-guided correction is simple and computationally efficient.

**Weaknesses:**

The claimed “condition-free” direct mapping from action to reaction is still conditioned on the action itself. The insight over conventional conditional diffusion models is unclear.

The demo video show mostly low-dynamic, stiff interactions with limited physical realism. Motions appear collision-free but lack natural dynamics and responsiveness expected in animation paper. This undermines the claimed advantage. I think the contribution is not enough if it just shows clear depenetration between humans, a post-hoc optimization applied after generation can achieve this easily as well.

Important baselines like ReMoS [1] are omitted. The reverse-generation results are minimal and not compelling.

[1] Ghosh, Anindita, et al. "Remos: 3d motion-conditioned reaction synthesis for two-person interactions." European Conference on Computer Vision. Cham: Springer Nature Switzerland, 2024.

**Questions:**

The proposed RE-GUID method adjusts the predicted clean pose (`x1_hat`) using a physical gradient and feeds the corrected result into the next sampling step, avoiding backpropagation through the network.

However, in principle, one could already do something similar in diffusion models:  `x1_hat' = x1_hat - λ * ∇ L_pene(x1_hat)` followed by `x_{t-1} = g(x_t, x1_hat')` where `g` is the usual reverse sampling update (e.g., DDPM or DDIM).

Could the authors clarify why RE-GUID is novel beyond this straightforward “detached guidance” idea? Is its contribution mainly the empirical integration into flow matching, or is there a theoretical reason why direct guidance on `x1_hat` would not work equivalently in diffusion models

---

> ### Author Response · Authors · 2025-11-27
> **Official Comment by Authors (1/2)**
>
> Thank you for the detailed review and thoughtful feedback. Please find our responses to your questions below.
> >Q1: The claimed “condition-free” direct mapping from action to reaction is still conditioned on the action itself. The insight over conventional conditional diffusion models is unclear.
>
> **A1:** In our Figure 1, actions and reactions are the two **endpoints** of our ODE trajectory, which is a simpler and more direct modeling approach, while the conventional diffusion is to learn a noise-to-reaction mapping conditioned on the action. Conventional conditional diffusion models typically fuse noise and conditional modalities through complex **cross attention** mechanisms and special designs for different tasks just as *"ReMoS"* you mentioned. During inference, the neural network needs to process input from the current state and conditions. However, due to our condition-free design, ARFlow **only** needs to process the input of the current state, which reduces the complexity of the network and can bring an additional benefit of bi-directional generation.
>
> >Q2-1: The demo video show mostly low-dynamic, stiff interactions with limited physical realism. Motions appear collision-free but lack natural dynamics and responsiveness expected in animation paper. This undermines the claimed advantage.
>
> **A2-1:** We thank the reviewer's observations. We would like to clarify that all of our previous videos were **at 0.5x speed**. The stiff interaction arises from our slow-speed processing of the output results, in order to more clearly demonstrate the advantages compared to SOTA methods. Limited physical realism is mainly due to the imperfections of fine-grained human-human interaction datasets currently available as stated in limitations of ReGenNet [1]. We have added **1x speed visualization results** in the updated supplementary video at **3:17**.
>
> >Q2-2: I think the contribution is not enough if it just shows clear depenetration between humans, a post-hoc optimization applied after generation can achieve this easily as well.
>
> **A2-2:** In the online setting, the action-reaction framework requires **real-time** performance where post-hoc optimization is **intolerable** and our proposed RE-GUID method is more accurate and efficient compared to traditional guidance methods. In addition, the initial point deviation problem we revealed is commonly present in the sampling of flow matching, and our modeling paradigm insights and RE-GUID method may have inspirational significance for other fields.
>
> >Q3: Important baselines like ReMoS [1] are omitted. The reverse-generation results are minimal and not compelling
>
> **A3:** ReMoS relies on intricate conditional mechanisms (combined spatio-temporal cross attention mechanisms) and uses a cascaded diffusion framework to generate more fine-grained reactive motions. However, it **cannot achieve online generation**. To explore the capability of this method in our task, we have supplemented its experiments as follows
>
> | Method   | FID↓  | Acc.↑ | Div.→ | MM.→  |
> | ---- | ----- | ----- | ----- | ----- |
> | Real     | 0.09  | 0.87  | 13.06 | 25.03 |
> | ReMoS    | 24.42 | 0.72  | 13.82 | 22.32 |
> | MDM-GRU  | 24.25 | 0.72  | 13.43 | 22.24 |
> | ReGenNet | 11.00 | **0.75**  | 13.80 | 22.90 |
> | **ARFlow**   | **8.07**  | 0.74  | **13.71** | **24.07** |
>
> We have added the above analysis and experimental results in **Appendix B.3** marked with blue color. We have also supplemented more experiments of reverse generation for all methods on NTU and Chi3D dataset in our revision.
> | Method   | FID↓  | Acc.↑ | Div.→ | MM.→  | Method   | FID↓  | Acc.↑ | Div.→ | MM.→  |
> | -------- | ----- | ----- | ----- | ----- |  -------- | ----- | ----- | ----- | ----- |
> | **Real-NTU**  | 0.01  | 0.59  | 16.01 | 25.78 | **Real-Chi3D**     | 0.80  | 0.60  | 7.17  | 13.42 |
> | cVAE     | 89.21 | 0.41  | 10.01 | 15.90 | cVAE     | 46.10 | 0.44  | 8.30  | 9.14  |
> | AGRoL    | 50.56 | 0.39  | 11.19 | 18.20 | AGRoL    | 67.90 | 0.49  | 6.21  | 8.89  |
> | MDM-GRU  | 42.13 | 0.42  | 12.23 | 20.33 | MDM-GRU  | 49.03 | 0.46  | 5.37  | 8.09  |
> | MDM      | 60.08 | 0.40  | 10.34 | 17.74 | MDM      | 48.20 | 0.50  | 5.89  | 8.41  |
> | ReGenNet | 36.12 | 0.46  | 12.66 | 19.30 | ReGenNet | 40.13 | 0.48  | 5.45  | 9.82  |
> | **ARFlow**   | **12.81** | **0.49**  | **14.84** | **23.40** |  **ARFlow**   | **13.89** | **0.55**  | **6.60**  | **12.03** |
>
> The experimental results further highlight the reverse generation capability of our method. In addition, we have added more reverse-generation **visualization results** in the updated supplementary video at **3:34**.
>
> **Reference:**
>
> [1] Liang Xu, et al. ReGenNet: Towards Human Action-Reaction Synthesis. CVPR, 2024.

---

> ### Author Response · Authors · 2025-11-27
> **Official Comment by Authors (2/2)**
>
> >Q4-1: The proposed RE-GUID method adjusts the predicted clean pose (x1_hat) using a physical gradient and feeds the corrected result into the next sampling step, avoiding backpropagation through the network.
> >However, in principle, one could already do something similar in diffusion models: x1_hat' = x1_hat - λ * ∇ L_pene(x1_hat) followed by x_{t-1} = g(x_t, x1_hat') where g is the usual reverse sampling update (e.g., DDPM or DDIM).
> >Could the authors clarify why RE-GUID is novel beyond this straightforward “detached guidance” idea? Is its contribution mainly the empirical integration into flow matching?
>
> **A4-1:** Thanks for your insightful comment. IF this straightforward "detached guidance" idea is used in flow matching framework, there will be **an issue** of *initial point deviation* that we have revealed. The point of flow matching reverse interpolation will deviate from the true initial point.
> Flow matching sampling actually interpolates back towards the predicted mean point of the source distribution instead of the true initial point, and the accumulating bias pulls the trajectory ever farther from the expected starting state. And this problem will not occur in diffusion models, because the source distribution of the diffusion model is a noise distribution, and its initial state is a noise **rather than** an initial action. To address the issue of initial point deviation, we propose to introduce a weighting factor $w$ in Eq.(14) to balance the predicted mean point and the true initial point. After adjusting the predicted clean pose $\hat{x}_{1}$ using a physical gradient, we project towards this revised point to correct the deviation (red directed solid line in Figure 3).
>
> >Q4-2: or is there a theoretical reason why direct guidance on x1_hat would not work equivalently in diffusion models?
>
> **A4-2:** Yes. The theoretical reason why direct guidance on x1_hat would not work equivalently in diffusion models is that it can lead to the problem of **manifold distortions** revealed by [1].
> [1] attempts to solve this problem by using an additional trained autoencoder to project the modified $\hat{x}_{1}$ onto a linear manifold. But this brings additional computational overhead, which is not suitable for our task that require high **real-time** performance.
>
> **Reference:**
>
> [1] Yutong He, et al. Manifold Preserving Guided Diffusion. ICLR, 2024.

---

### Official Review · Reviewer_husG · 2025-11-01

**Soundness:** 3
**Presentation:** 3
**Contribution:** 3
**Rating:** 6
**Confidence:** 4

**Summary:**

This paper addresses the challenge of human action-reaction synthesis, which aims to generate physically plausible human reactions in response to given actions. The authors identify two key limitations in existing method of 1. Complex and unidirectional generation; 2. Physical violations. To solve these problems, this paper proposes Action-Reaction Flow matching (ARFlow), a novel framework built on flow matching, to directly model the mapping from the action distribution to the reaction distribution. The simple architecture supports bi-directional generation for reaction-to-action generation. Besides, the authors identify the issue of “Initial Point Deviation” for physical violations of human-human interactions. Then, RE-GUID is proposed as a reprojection guidance to prevent penetration. Extensive experiments on NTU120, Chi3D and InterHuman show the superiority of ARFlow.

**Strengths:**

1. This work is the first work to apply a flow matching framework to the human action-reaction synthesis task. This "condition-free" approach, which directly models the mapping from the action distribution to the reaction distribution, is an elegant departure from existing diffusion models that rely on complex conditional mechanisms. Critically, this design choice directly enables bi-directional generation (both action-to-reaction and reaction-to-action), solving a key limitation of prior unidirectional methods.
2. The paper's method for handling body inter-penetration is highly insightful. The proposed RE-GUID method is an effective and efficient solution that corrects this deviation by re-projecting the interpolation endpoint . This method, validated by the paper's newly introduced IF and IV metrics, is shown to dramatically reduce body collisions far below the levels of prior work.
3. The authors provide thorough experimental validation across multiple challenging datasets (NTU120-AS, Chi3D-AS, and InterHuman-AS). The results are compelling, demonstrating that ARFlow outperforms existing state-of-the-art methods on key metrics like Fréchet Inception Distance (FID) and motion diversity. Furthermore, the proposed model is significantly more efficient, with fewer parameters, faster training convergence (e.g., half the time of ReGenNet), and much lower inference latency.
4. The paper is well-written and organized. The authors clearly motivate the problem, articulate the limitations of existing work, and present their technical contributions in a logical, easy-to-follow manner.

**Weaknesses:**

1. Limited Extensibility to Conditional Generation: The paper's primary innovation is its "condition-free" design. The paper provides no experiments or results for the conditional setting, making its flexibility and practical utility in constrained scenarios unproven.
2. Unclear Bi-Directionality for Dynamic Interactions: The claim of "bi-directional generation" is a key strength, but its practical application seems limited. Besides, for the “boxing” example, it is unclear how the current framework could support such a continuous, auto-regressive switching of roles, as this would require a different model setup not demonstrated in the paper.
3. Insufficient Evidence for Reaction Diversity: The model claims to support diverse reaction motions for the same action by incorporating stochasticity during sampling. The paper lacks clear visualization examples for diverse reaction generation. Without this, it is difficult to assess if the model is generating truly meaningful variations or just minor, low-impact perturbations of the same motion pattern.
4. Limited Generalizability of Physical Constraints: The RE-GUID method, while effective at reducing penetration, may not generalize well. From the visualization, current generated results still physically implausible.

**Questions:**

The authors are asked to clarify:
1. **Conditional Generation:** How can the "condition-free" model be extended to handle conditional inputs like text, since this was not demonstrated?
2. **Dynamic Bi-Directionality:** How does the *static* "reaction-to-action" experiment support the claim of handling *dynamic*, role-switching interactions (e.g., "boxing") ?
3. **Visual Evidence:** Please provide more visual results to prove:
    - **Diversity:** Show multiple, *distinct* reactions to the *same* action.
    - **Physical Realism:** Show that the model works for *close-contact* interactions (like handshakes) and doesn't just "unnaturally avoid" contact .

---

> ### Author Response · Authors · 2025-11-27
>
> Thank you for the detailed review and thoughtful feedback. Please find our responses to your questions below.
> >Q1:Conditional Generation: How can the "condition-free" model be extended to handle conditional inputs like text, since this was not demonstrated?
>
> **A1:** Our model is condition-free for this task (online, unconstrained setting as stated in Line.311-312), but it can be extended to handle additional conditional input $c$ at the dashed line in the left column of our Figure 2 by directly incorporating $c$ into the vector field estimator (Line.170-175). We follow your suggestion to test our method on the constrained settings (text serves as conditional input).
>
> | Method               | FID↓ | Acc.↑ | Div.→ | MM.→  |
> | -------------------- | ---- | ----- | ----- | ----- |
> | Real                 | 0.09 | 0.87  | 13.06 | 25.03 |
> | ARFlow-unconstrained | 8.07 | 0.74  | 13.71 | 24.07 |
> | ARFlow-constrained   | 4.56 | 0.89  | 12.00 | 25.06 |
>
> This setting achieves superior performance since the text serves as a strong hint to generate the reactions. We have added the above analysis and experimental results in **Appendix B.4** marked with blue color.
>
> >Q2: Dynamic Bi-Directionality: How does the static "reaction-to-action" experiment support the claim of handling dynamic, role-switching interactions (e.g., "boxing")?
>
> **A2:**  We follow your suggestion to test our method on the role-switching interactions on InterHuman-AS dataset. To handling dynamic, role-switching interactions, we introduce a pre-trained classifier to distinguish the current state of input motion ("Actor" or "Reactor"). During inference, ARFlow determines whether to perform forward or reverse generation based on the current state output by the classifier.
>
> | Method   | R Precision↑ | FID↓ | MM Dist.↓ | Div.→ | MM.→ |
> | -------- | :------: | :------: | :------: | :------: | :------: |
> | Real     | 0.75         | 0.01  | 3.42      | 6.31  |   -   |
> | MDM      | 0.34         | 15.10 | 11.55     | 5.20  | 1.98 |
> | ReGenNet | 0.36         | 12.19 | 10.38     | 5.89  | 2.09 |
> | **ARFlow**   | **0.42**         | **3.89**  | **4.26**      | **6.03**  | **2.29** |
>
> The experimental results show that ARFlow has significantly better performance than the diffusion-based methods due to the stronger performance of our model in reverse generation. More details are referred to **Appendix B.5** marked with blue color please.
>
> >Q3: Diversity: Show multiple, distinct reactions to the same action. The paper lacks clear visualization examples for diverse reaction generation. Without this, it is difficult to assess if the model is generating truly meaningful variations or just minor, low-impact perturbations of the same motion pattern.
>
> **A3:** We have included more visual results in the updated supplementary video at **3:46** to demonstrate our model's ability to generate truly meaningful variations. To generate reactions with larger variations in motion patterns for the same action, it relies on larger and more diverse training data. Because models struggle to generate reactions that are significantly different from those seen during training.
>
> >Q4: Physical Realism: Show that the model works for close-contact interactions (like handshakes) and doesn't just "unnaturally avoid" contact. Limited Generalizability of RE-GUID method to other physical constraints.
>
> **A4:** Thanks for your valuable feedback. The close-contact interaction is indeed a limitation of our current method, which we have acknowledged in our Limitations (Line. 489) and analyzed the failure case of handshaking in our **Figure J.2** of Appendix J. In the revision, we have supplemented the more systematic analysis of this failure case (Line.1217-1224) marked with blue.
>
> As for the issue of generalization, we want to highlight that our RE-GUID method is **compatible with** various fine-grained loss functions (contact loss [1], penetration loss [2], foot-skating loss [3,4]) which can be seamlessly integrated into our method to handle different types of physical constraints. In the revision, we have added the foot-skating loss to generate more physically plausible motions. More details are referred to **Appendix D.4** marked with blue color.
>
> **Reference:**
>
> [1] Jie Tian, et al. Gaze-guided Hand-Object Interaction Synthesis: Dataset and Method. arXiv e-prints, pp. arXiv–2403, 2024.
>
> [2] Jihyun Lee, et al. InterHandGen: Two-Hand Interaction Generation via Cascaded Reverse Diffusion. CVPR, 2024.
>
> [3] Yuliang Zou, et al. Reducing Footskate in Human Motion Reconstruction with Ground Contact Constraints. CVPR, 2020.
>
> [4] Ye Yuan, et al. Physdiff: Physics-guided Human Motion Diffusion Model. ICCV, 2023.

---

### Author Response · Authors · 2025-12-03
**Final Remark**

Dear AC,

We sincerely appreciate the considerable effort you dedicated to reviewing our manuscript under these special circumstances. For your reference, we would like to briefly summarize our discussions with the reviewers so far and how we addressed these concerns.

We sincerely thank all the reviewers for their valuable feedback and constructive comments. We are encouraged by the reviewers' positive feedback, noting that our paper is the **first work**
 to apply a flow matching framework to the task (R-husG), our proposed paradigm is elegant (R-husG), novel (R-fcJX) and interesting alternative (R-EMJ3,R-6px3) to diffusion, our proposed RE-GUID method is highly insightful (R-husG), effective (R-fcJX) and efficient (R-EMJ3,R-6px3) to reduce body penetration, and our experiments are thorough (R-husG), extensive (R-fcJX) with better performance (R-6px3). The reviewers also highlighted the Initial Point Deviation issue (R-fcJX) we identified and the new metrics should be of interest to the community (R-6px3). *All* the reviewers agree with our method's faster inference ability.

---
 **1. Reviewers husG and fcJX maintain positive scores** and the latter replied that we have addressed his questions.

----
**2. Reviewer 6px3** expressed a tendency to improve scores with only one issue requires further clarification. We have added comparisons of two recent baselines (Ready-to-React, ReMos), experiments on constrained settings and **experimental and visualization** results for handling **role-switching interactions**, as he suggested. **His remaining questions** are about the details and visualization results of our supplementary role-switching experiments. We have provided detailed responses and **visualization** results. The above evidence further shows our method's superior performance and strengthens our paper's central claim regarding **bi-directional generation ability to handle role-switching interactions**.

----
**3. Reviewer EMJ3** cannot reply due to the closed system and his **key** concerns are as follows

> The insight over conventional conditional diffusion models is unclear.

Actions and reactions are the two **endpoints** of our ODE trajectory, which is a simpler and more direct modeling approach, while the conventional diffusion is to learn a noise-to-reaction mapping conditioned on the action through complex conditional mechanisms(e.g. cross attention) where the neural network needs to process **extra input** from the conditions.

> The demo video show mostly low-dynamic, stiff interactions with limited physical realism. The reverse-generation results are minimal and not compelling.

This phenomenon is because our original video was **0.5× Speed** to show more details. During the rebuttal, we added the **1.0× Speed** video results as well and also supplemented more **reverse-generation** experimental and visualization results to further highlight our method's reverse generation capability.

> Why RE-GUID is better than this straightforward "detached guidance" idea?

Directly using this straightforward "detached guidance" idea will lead to the **issue** of *initial point deviation* we revealed. We propose to introduce a weighting factor to balance the predicted mean point and the true initial point, projecting towards this revised point to correct the deviation.

---
Finally, thank you again for all your valuable time and kind consideration!

Best regards,

The Authors

---

### Meta-Review · Area_Chair_RDcf · 2026-01-07

**Summary:**

Across the reviews and discussion, the decision mainly hinges on whether ARFlow’s conceptual novelty translates into competitive motion quality and realism. The central remaining worry is perceived output quality. The method of flow matching for direct action to reaction mapping and guidance integrated into flow sampling is found elegant and fast, but the generated motions look stiffer and less physically “alive” than baselines in closely related settings, e.g., ReMoS and methods like InterMask, which make the contribution narrow despite being technically interesting.

**Reviewer Concerns:**

**Addressed in the rebuttal or revision:**

- Missing baselines
- Role-switching and bidirectionality claims
- Text-conditioned extension
- “Detached guidance” vs. RE-GUID novelty
- Metric/table omissions and clarity issues
- Failure case acknowledgment, e.g., handshake/contact precision

**Still not fully resolved:**
- Qualitative motion realism. Many examples show foot floating (noticeable in 3:53-3:58). The issue can be partly due to the dataset and baseline, but existing work shows more physically “alive” motion, e.g., ReMoS and closely related tasks like InterMask.

**Reviewer Scores:**

husG likely stays at 6: Their main concerns, including conditional setting, diversity visuals, and role-switching evidence, were addressed; however, some limitations remain.

EMJ3 likely stays at 4: while the rebuttal addresses omissions like the ReMoS baseline and reverse generation results, and clarifies RE-GUID’s flow-specific novelty, their biggest concern was motion realism, which seems not on par with the existing baselines and related work showing more believing and life-like animation.

fcJX stays at 6 as explicitly stated.

6px3 likely moves from 4 to higher score. The authors answered baselines, role-switching, conditioned setting details, and qualitative videos; the reviewer provided follow-ups. The remaining includes doubts about reliance on role annotation, which has been addressed in the responses.

---

### Decision · Program_Chairs · 2026-01-26

Reject